**Resource**

# Single-nucleus RNA-seq dissection of choroid plexus tumor cell heterogeneity

Anthony D Hill ⓘD [1✉], Konstantin Okonechnikov[2], Marla K Herr[1,4], Christian Thomas[3], Supat Thongjuea[2], Martin Hasselblatt[3] & Annarita Patrizi ⓘD [1✉]

## Abstract

**The genomic, genetic and cellular events regulating the onset, growth and survival of rare, choroid plexus neoplasms remain poorly understood. Here, we examine the heterogeneity of human choroid plexus tumors by single-nucleus transcriptome analysis of 23,906 cells from four disease-free choroid plexus and eleven choroid plexus tumors. The resulting expression atlas profiles cellular and transcriptional diversity, copy number alterations, and cell–cell interaction networks in normal and cancerous choroid plexus. In choroid plexus tumor epithelial cells, we observe transcriptional changes that correlate with genome-wide methylation profiles. We further characterize tumor type-specific stromal microenvironments that include altered macrophage and mesenchymal cell states, as well as changes in extracellular matrix components. This first single-cell dataset resource from such scarce samples should be valuable for divising therapies against these little-studied neoplasms.**

**Keywords** Choroid Plexus Tumors; Single Cell RNAseq; Tumor Microenvironment; Cerebrospinal Fluid (CSF)
**Subject Categories** Cancer; Methods & Resources; Neuroscience

## Introduction

Choroid plexus tumors (CPT) are rare, usually pediatric neoplasms arising from the epithelial cells of the choroid plexus (ChP). While rare, these tumors represent 10–20% of central nervous system (CNS) neoplasms during the first year of life (Cannon et al, 2015; Ostrom et al, 2014; Rickert and Paulus, 2001). CPT can be histologically distinguished as choroid plexus papilloma (CPP, WHO grade 1), atypical CPP (aCPP, WHO grade 2), and choroid plexus carcinoma (CPC, WHO grade 3). While patients with CPP generally have excellent outcomes following surgery, patients with CPC have high rates of recurrence and poor (≈50%) 5-year survival

rates (Cannon et al, 2015). More recently, DNA methylation profiles have been used to segregate CPTs into three clinically relevant subgroups (Capper et al, 2018; Merino et al, 2015; Pienkowska et al, 2019; Thomas et al, 2016). Methylation cluster 1 ("pedA") comprises CPPs and aCPPs characterized by young age and mainly supratentorial location, whereas methylation cluster 2 ("adult") contains infratentorial CPP and aCPPs of adult patients. Tumors of both methylation clusters 1 and 2 show favorable prognosis. In contrast, all CPC, but also a considerable number of pediatric CPP and aCPP, are assigned to high-risk methylation cluster 3 ("pedB") exhibiting frequent recurrences.

The past quarter century has shed some light on the genetics of CPT. CPC are among several tumor types that occur in Li-Fraumeni families harboring germline mutations of *TP53*, and somatic *TP53* mutations have been found in about half of all choroid plexus carcinomas (Kamaly-Asl et al, 2006; Tabori et al, 2010; Thomas et al, 2021). *TERT* promoter hypermethylation and mutations have also been identified in a subset of CPT cases (Castelo-Branco et al, 2013; Thomas et al, 2021). Overexpression of *Myc* in the murine CNS, alone or in combination with *Tp53* deletion, causes CPT in transgenic mice (El Nagar et al, 2018; Merve et al, 2019; Shannon et al, 2018; Wang et al, 2019). A few additional genes, including *Notch1*, and *Pten* have also been associated with tumors of the choroid plexus (ChP) in mice (Li et al, 2022). A search for syntenic regions amplified in human and mouse CPC identified *TAF12*, *NFYC*, and *RAD54L* as oncogenes required for disease initiation and progression (Tong et al, 2015). However, no common driver mutations have been identified for CPT, and the cellular and molecular events leading up to the initiation, growth and immune evasion of these tumors remain largely unknown.

The healthy ChP acts as a barrier between the blood and cerebral spinal fluid (CSF), site of CSF production, source of signals controlling brain development and maturation, and interface between the brain and immune system (Kaiser et al, 2019; Lehtinen et al, 2011; Stolp et al, 2013). This broad and dynamic set of functions is made possible by the orchestrated production, recruitment, and interaction of multiple cell types, including epithelial, mesenchymal, endothelial, and immune cells (Lun et al, 2015). How the multiple types of ChP intrinsic, as well as

[1]Schaller Research Group, German Cancer Research Center (DKFZ), 69120 Heidelberg, Germany. [2]Division of Pediatric Neurooncology, German Cancer Research Center (DKFZ) and German Cancer Consortium (DKTK), 69120 Heidelberg, Germany. [3]Institute of Neuropathology, University Hospital Münster, 48149 Münster, Germany. [4]Present address: Division of Molecular Neurobiology, Department of Medical Biochemistry and Biophysics, Karolinska Institute, 17177 Stockholm, Sweden. ✉E-mail: a.hill@dkfz.de; a.patrizi@dkfz.de

recruited extrinsic cells modulate ChP function across development and in response to aging and disease are being actively studied by a growing number of labs. However, essentially nothing is known about how these interactions are perturbed in CPT, how non-epithelial cell types might contribute to the growth and survival of CPT cells, or the effect of CPT cells on immune cell function.

As a first step towards profiling the cell types, gene expression patterns and cellular interactions within CPT, we performed droplet-based RNAseq on nuclei from four disease-free ChP samples, three adult CPT and eight pedB CPT. We observed significant differences in gene expression, copy number variation (CNV) and barrier/transport function among ChP and two types of CPT. In addition, we characterized an altered tumor microenvironment, including tumor type-specific extracellular matrix gene expression and heterogeneity in stromal and immune cell states among the tumor types. Taken together these results constitute a cellular atlas of choroid plexus tumors, and highlight major pathways altered in each tumor group.

# Results

## Cellular composition in human disease-free choroid plexus and choroid plexus tumors

We used 10xGenomics droplet-based sequencing to profile single nuclei RNA gene expression in 23,906 cells from 15 human ChP and CPT samples (Table 1). Comparison of the mapped read ratio and mitochondrial reads by sample type and methylation tumor class showed similarity with a previously reported dataset (Yang et al, 2021) (Appendix Table S1). CPT were categorized in two groups based on their methylation profile, adult and pedB. The adult CPT group contained three papilloma samples (CP1, CP3, and CP4) characterized by infratentorial low-risk CPTs, while the pedB group contained all remaining tumor samples with supratentorial pediatric high-risk CPTs (Table 1 and Appendix Table S1). Clustering and dimensionality reduction of the dataset revealed considerable heterogeneity among the cells, which form 14 clusters (Fig. 1A). Interestingly, the cells cluster into large overlapping groups by methylation profile: pedB (green), ChP (gold), and adult (blue) (Fig. 1B). By using marker gene expression to identify the cell types present in the 14 clusters, we noticed that markers for non-epithelial cells (endothelial, mesenchymal, macrophage and neural cells, as well as Tcells) were each found to be expressed in a single cluster with contributions from most if not all samples and all four sample types (Fig. 1C,D). The remaining 9 clusters express the choroid plexus epithelial cell lineage markers *OTX2* and *LMX1A*. One of these 9 *OTX2*$^+$/*LMX1A*$^+$ clusters robustly express markers of mature choroid plexus epithelial cells (CPECs), including *OTX2* (Dani et al, 2021), *SLC4A10* (Christensen et al, 2013), and *SLC39A12* (Ho et al, 2012). This cluster, called ChP-Epi, is mainly composed of cells from all four disease-free samples, as well as aCP2. Disease-free ChP cells were excluded from a pair of *OTX2*$^+$/*LMX1A*$^+$ clusters labeled "CC/CP-Epi" and CP/CC-Epi, comprised of cells from samples CP1, CP2, CC1, CC3, and CC4 in different ratios. The remaining six clusters, each of which primarily contains cells from one or two tumor samples, expressed varying levels of additional CPEC markers. Epithelial cells from adult profile samples, mainly found in the clusters CP1/3-Epi and CP4-

Epi, express higher levels of CPEC markers than the pedB cells found in clusters aCP1-Epi, CP2-Epi, CC2-Epi, CC3-Epi, and CC/CP-Epi (Fig. 1D). In brief, CPEC cells from the disease-free ChP samples are excluded from CPT clusters. Among tumor samples, cells of an epithelial lineage appear to be more heterogeneous than non-epithelial cells, and pedB cells appear to be more divergent from CPECs than adult cells.

In order to visualize the constituent cell types in disease-free ChP, adult CPT and pedB CPT, we created methylation profile specific integrated data sets by using the batch-correction and data integration tools available in Seurat (Stuart et al, 2019). Consistent with previous findings for human (Yang et al, 2021) and mouse (Dani et al, 2021) ChP, we observed epithelial, endothelial, mesenchymal, macrophage, neuronal, glial and T-cell marker expressing clusters in our disease-free human ChP dataset (Fig. 2A). In addition, we observed a previously unreported diversity of human ChP mesenchymal cells, with three clusters expressing ChP mesenchymal cell markers including *COL1A2*, *COL3A1*, and *ADAM12* (Fig. EV1A). One cluster was designated mesenchymal-PC, because cells in this cluster express classically defined pericyte markers including *PDGFRB*, *RGS5*, and *ANGPT1* (Fig. EV1A). Two additional clusters, designated as mesenchymal-1 and mesenchymal-2, expressed overlapping but distinct subsets of mesenchymal/fibroblast marker genes (Fig. EV1A). Similarly, we detect two clusters expressing endothelial markers such as *PECAM1*, *VWF,* and *FLT* (Schupp et al, 2021) (Fig. EV1B). One cluster (called "Endothelial") also expresses markers of fenestrated capillaries (*PLVAP*) and venous endothelial cells (*TSHZ2*), while the second expresses markers of vascular smooth muscle cells (*MYH11*, *ACTA2*, and *CARMN*) (Fig. EV1B). Neuronal (*NRXN1, DCLK1, NRCAM*) and glial (*FMN2, ERBB4, NPAS3*) marker expressing clusters were also captured in this dataset (Fig. EV1C).

In the adult CPT profile dataset, we detected epithelial, endothelial, mesenchymal, and immune cell marker expressing clusters (Fig. 2B). A similar set of clusters were found in the pedB CPT (Fig. 2C), with two exceptions: (i) we did not detect a mesenchymal cell cluster in the pedB dataset and (ii) the presence of three cell clusters expressing CPEC lineage marker *OTX2* and *LMX1A*, here identified as "Epithelial1", "Epithelial2", and "Proliferative" (Fig. 2D). Cells within epithelial1 cluster express disease-free CPEC markers identified in Fig. 1. Cells in the epithelial2 cluster show little expression of these markers, but increased expression of genes including *NEAT1*, *PKM*, *STRA6*, and *KCNQ1OT1*. Cells within the proliferative cluster express CPEC lineage markers, epithelial2 markers, as well as markers of proliferating cells, including *ASPM*, *TOP2A*, *CENPF*, and *MKI67* (Fig. 2D). No neuronal expressing clusters were observed in either tumor profile. Taken together, we confirmed the major cell types of the human ChP in disease-free ChP and CPT, in addition to a tumor-specific proliferative cell type. Batch correcting ChP and CPT data by sample type highlights heterogeneity among ChP mesenchymal and vascular cell types, as well as CPT epithelial cell types.

In order to compare gene expression between disease-free ChP and CPTs, we performed data integration on the entire single nucleus dataset. After clustering and dimensionality reduction, cells formed six clusters corresponding to the broad cell types observed previously (Fig. EV2A,B). Quantitation of relative cell numbers per sample type revealed a shift in the composition of CPT samples relative to ChP. Fewer mesenchymal cells were observed in the adult and pedB populations relative to ChP. Instead, the tumor populations contained

**Table 1. Detailed information of fresh-frozen disease-free choroid plexus (ChP) and choroid plexus tumor samples.**

| Sample | Diagnosis | Age | Gender | Location | Methylation (v11b4) | Calibrated score (v11b4) | Methylation (v12.8) | Calibrated score (v12.8) |
|---|---|---|---|---|---|---|---|---|
| ChP1 | - | 78 | M | LV | - | - | - | - |
| ChP2 | - | 47 | F | LV | - | - | - | - |
| ChP3 | - | 24 | F | | - | - | - | - |
| ChP4 | - | 24 | F | | - | - | - | - |
| CP1 | CPP | 72 | M | 4V | Adult | 0.17215 | No methylation class with score >0.1 | - |
| CP2 | CPP | 0.9 | M | LV | Pediatric B | 0.99909 | Choroid plexus carcinoma, pediatric subtype | 0.99822 |
| CP3 | CPP | 30 | F | - | Adult | 0.99948 | Choroid plexus papilloma, adult subtype | 0.99999 |
| CP4 | CPP | 47 | M | - | Adult | 0.99995 | Choroid plexus papilloma, adult subtype | 0.99999 |
| aCP1 | aCPP | 5 | M | LV | Pediatric B | 0.99999 | Choroid plexus carcinoma, pediatric subtype | 0.99998 |
| aCP2 | aCPP | 12 | F | LV | Pediatric B | 0.96573 | Choroid plexus carcinoma, adult subtype | 0.97557 |
| CC1 | CPC | 0.8 | F | Parietal left | Pediatric B | 0.99997 | Choroid plexus carcinoma, pediatric subtype | 0.99998 |
| CC2 | CPC | 0.6 | F | LV | Pediatric B | 0.99999 | Choroid plexus carcinoma, pediatric subtype | 0.99999 |
| CC3 | CPC | 30 | M | LV | Pediatric B | 0.98975 | Choroid plexus carcinoma, adult subtype | 0.99999 |
| CC4 | CPC | 16 | M | - | Pediatric B | 0.99970 | Choroid plexus carcinoma, pediatric subtype | 0.99953 |
| CC5 | CPC | 6 | M | - | Pediatric B | | Choroid plexus carcinoma, pediatric subtype | 0.98541 |

Diagnosis, age, gender, location, and methylation profile for samples in this study (disease-free choroid plexus = ChP, CPP = choroid plexus papilloma, aCPP = atypical choroid plexus papilloma, CPC = choroid plexus carcinoma, M = male, F = female, LV = lateral ventricle, 4V = fourth ventricle).

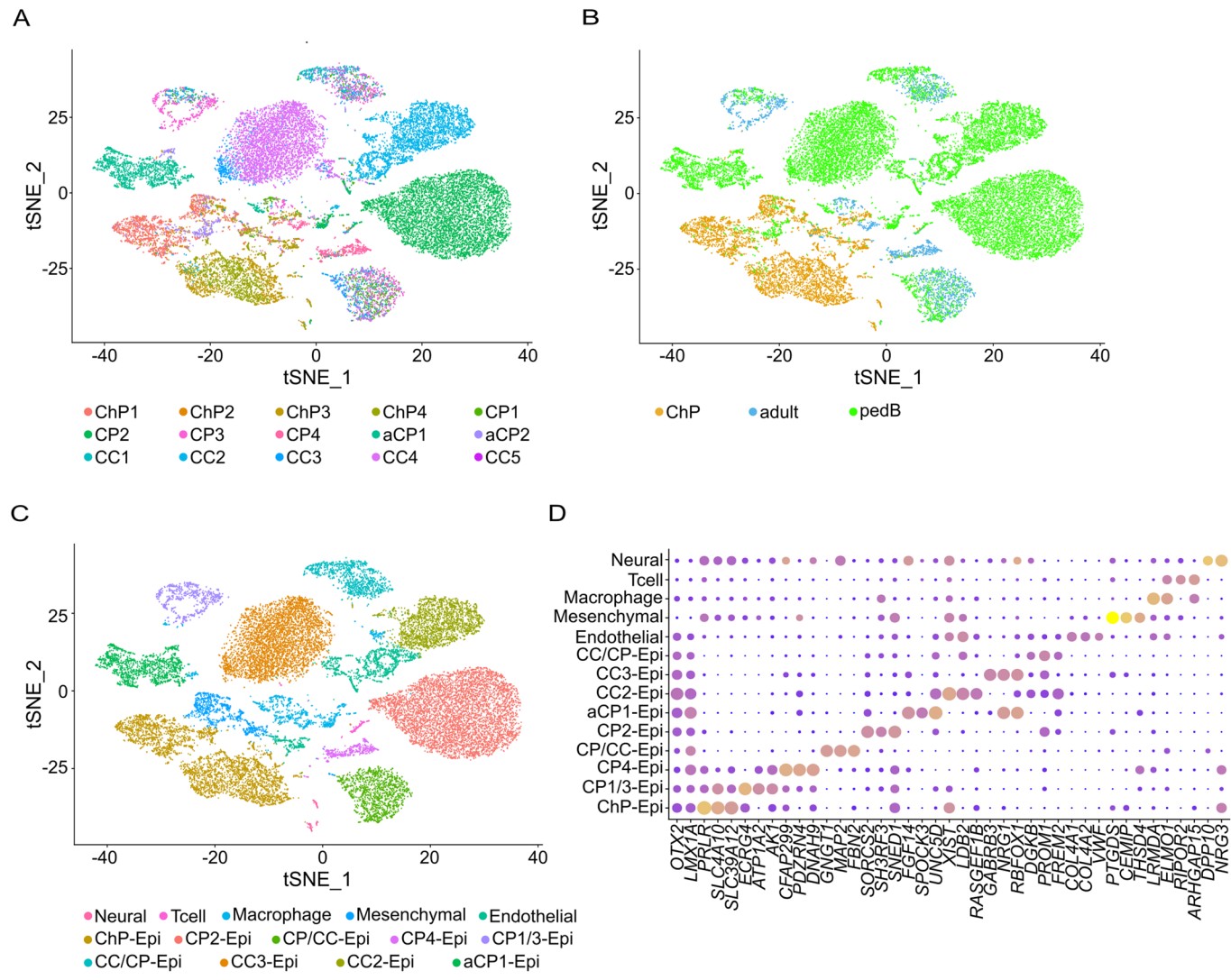

**Figure 1. Cellular diversity of disease-free choroid plexus and choroid plexus tumor samples reflect tumor type and donor age.**

(A) Graph-based clustering of human choroid plexus tumors colored by sample. (B) Identical clustering in (A) colored by methylation profile, green = pediatric high-risk choroid plexus tumors (pedB), cyan = adult low-risk choroid plexus tumors (adult), gold = disease-free choroid plexus. (C) Clustered cells colored by cell type. (D) Dotplot of lowest adjusted *p*-value cell type-specific markers used to assign cell identity.

more macrophage and proliferative cells (Fig. EV2C). We next analyzed the cellular composition of individual ChP and CPT samples, grouped by methylation profile (Fig. EV2D). The percentage of total macrophage was significantly higher in adult vs. pedB CPT samples (average percent macrophage per sample 22.4% adult vs. 8.2% pedB, adjusted *p*-vale 0.046). Both adult and pedB showed a reduction of the percent of total mesenchymal cells compared to ChP (average percent mesenchymal cells 15.8% ChP, 3.3% adult tumors, 5.1% pedB tumors), however, only pedB reached significance (adjusted *p*-value 0.029). Finally, both adult and pedB CPT contained an increased percentage of proliferative cells, with only pedB reaching statistical significance compared to ChP (0.2% ChP vs. 3.7% pedB tumors, adjusted *p*-value 0.016). We next used the cell cycle scoring tools present in Seurat to characterize cell proliferation in CPT. Consistent with the risk profile of pedB tumors, cells in pedB samples were far more likely to score as S or G2/M phase (Fig. EV2E). Moreover, the proliferative cluster

appears to be entirely comprised of cells in S or G2/M phase. Integration of the entire dataset revealed shifts in tumor cell composition relative to disease-free ChP and confirmed the hyper-proliferation of a subset of pedB cells.

## Genomic and gene expression changes in human choroid plexus tumors

Frequent copy number changes have been observed in bulk human CPT (Merino et al, 2015; Thomas et al, 2021), as well as in a murine model of CPC (Wang et al, 2019). We hypothesized that the gain and loss of chromosome segments could play important roles in the growth and evolution of CPT in vivo, and that profiling copy number at single nucleus resolution might give insights into these events. We used InferCNV to identify variations in copy number across our dataset. Not surprisingly, the broad pattern of

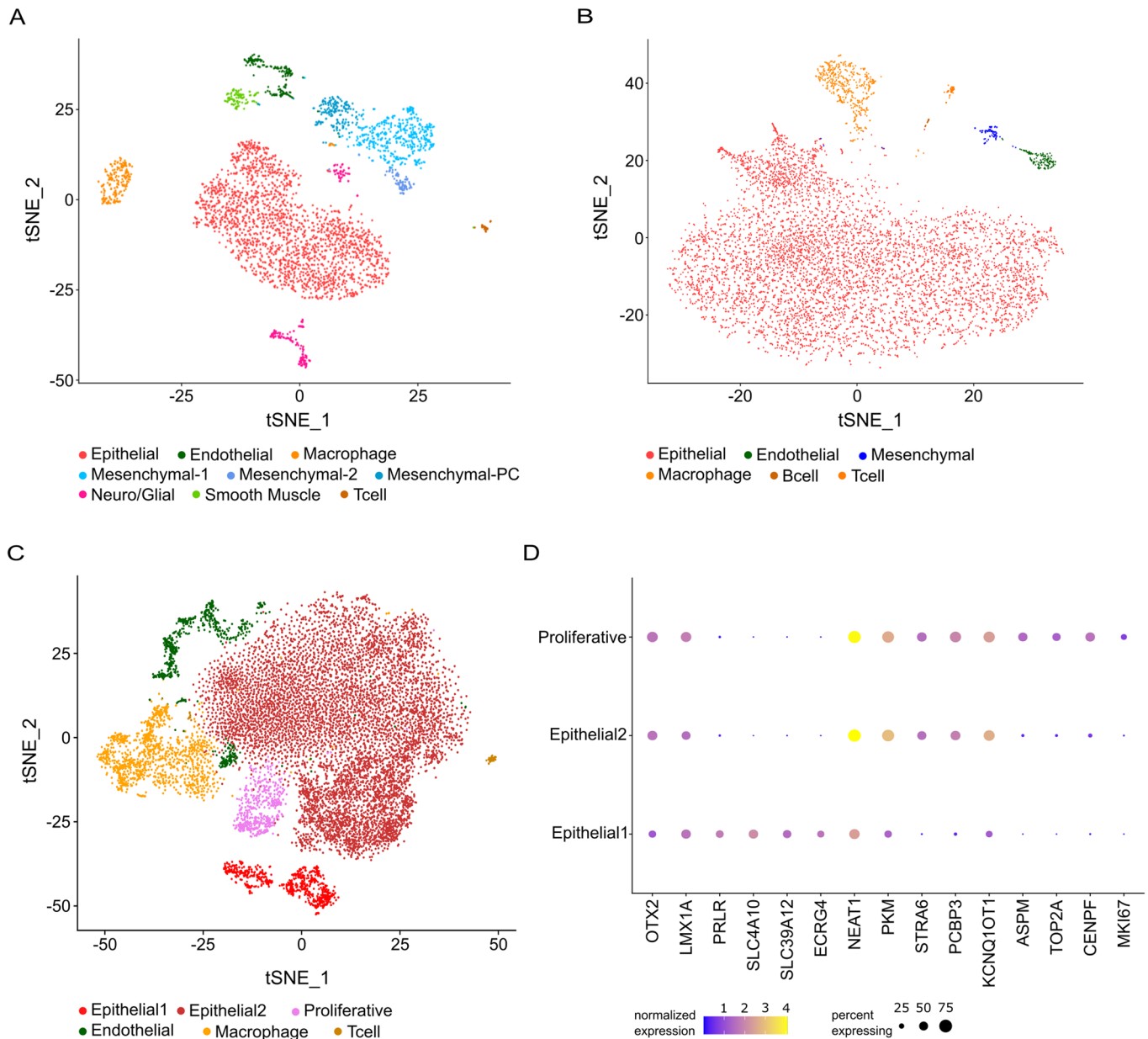

**Figure 2. Constituent cell types of disease-free choroid plexus (ChP) and choroid plexus tumors (CPTs).**

(A) tSNE of batch-corrected ChP snRNAseq data colored to reflect major epithelial, immune, vascular, and neural cell types (4732 nuclei). (B, C) tSNE of batch corrected cells from adult profile CPT (B) and pedB profile CPT (C) colored to indicate the same broad cell types as in (A) above, as well as a tumor-specific proliferative (magenta) cluster (1313 and 17,861 nuclei, respectively). (D) Dotplot showing marker gene expression in three epithelial lineage (*OTX2*$^+$ and *LMX1A*$^+$) clusters from (C). Average expression level in epithelial cells (color) and proportion of expressing cells (circle size) of selected candidate genes (column) identified in Fig. 1D. Candidate genes from epithelial2 and proliferative were selected based on lowest adjusted *p* value.

chromosome arm gain and loss differed among samples (Fig. 3A). However, some patterns were evident: the most common CNVs in adult profile tumors involved gain of 9p and 9q as well as loss of 21p and 21q. Common CNVs in pedB tumors include gain of 1p, 1q, 4p, 4q, 12p, 12q, and 20p (Fig. 3B), partially consistent with results obtained from larger cohorts (Thomas et al, 2021).

We next compared gene expression in epithelial and proliferative cells from ChP, adult, and pedB samples using pseudobulked raw counts for each sample. Principal component analysis (PCA) and

hierarchical clustering of gene expression segregate tumor samples by methylation profile, tumor grade, and patient age (Fig. 4A). Samples with a pedB methylation profile, comprising all CPC samples in the study plus one CPP sample (CP2), form one group, while adult CPT methylation profile samples (CP1, 3 and 4) were contained in a second distinct group. Within the pedB group, neonate samples possessed the most similar gene expression, while that of the sole pedB sample from an adult patient, CC3, showed the least similarity. Both tumor groups are distinct from disease-free ChP.

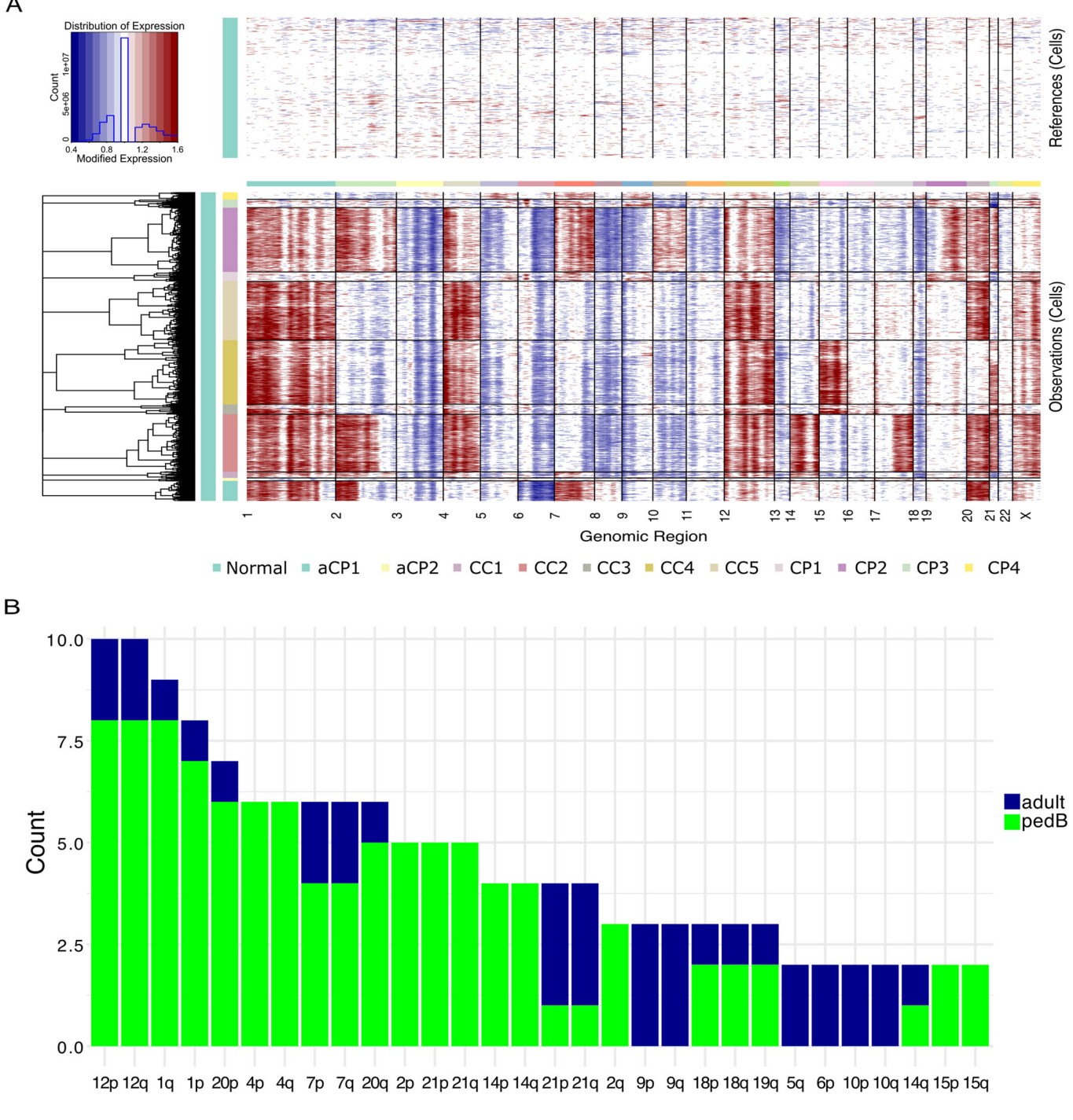

**Figure 3. Copy number variation (CNV) in choroid plexus tumor (CPT).**

(**A**) Copy number variance profiling per sample performed with InferCnv toolkit across single nucleus for all CPT cases (bottom panel, 19,435 cells) using combined disease-free choroid plexus (ChP) as reference (top panel, 4795 cells). (**B**) Barplot showing number of samples with a given CNV. Histogram: green = pedB, blue = adult; bottom bar: red = gain, cyan = loss.

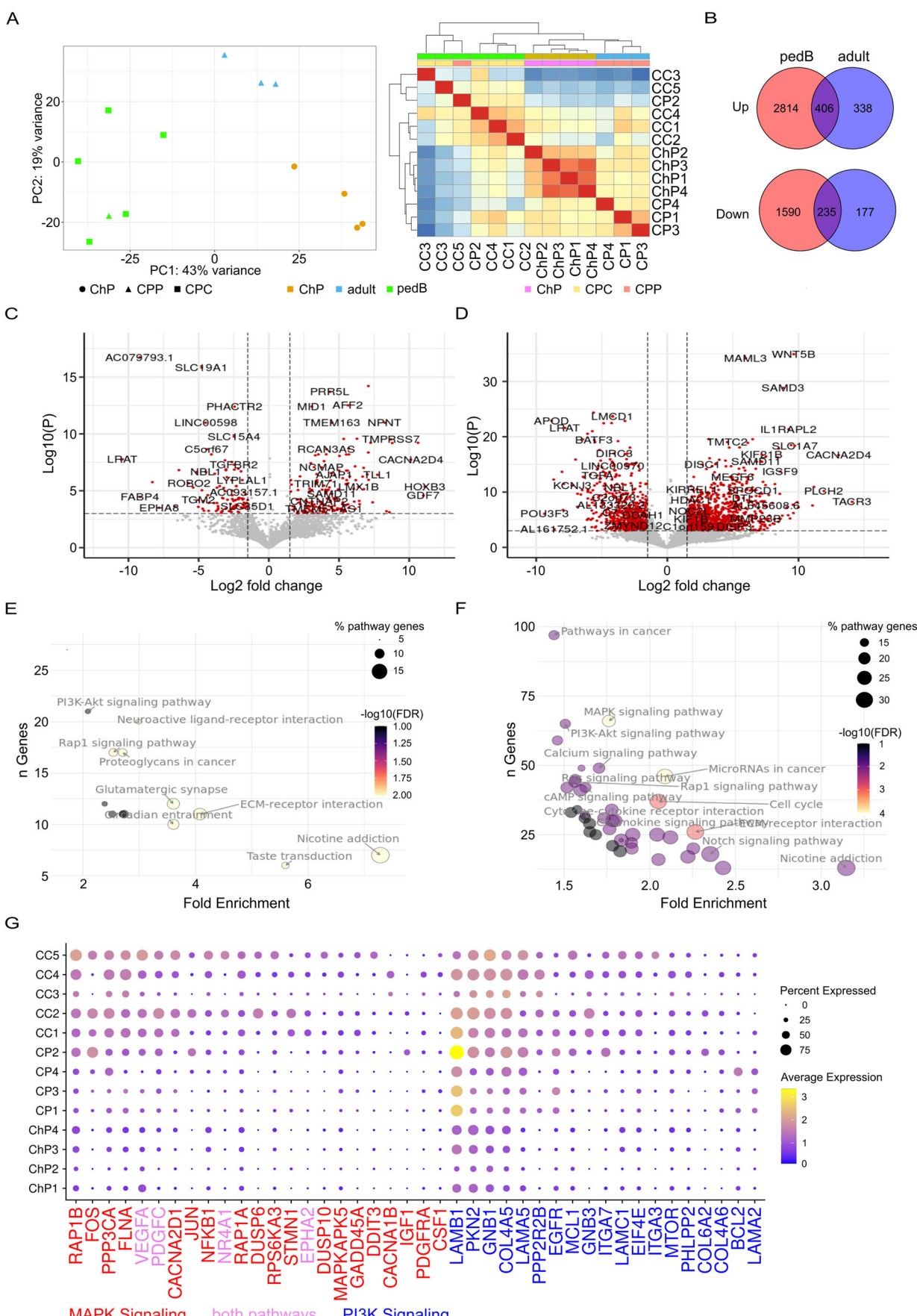

◄  **Figure 4.   Differential gene expression in choroid plexus tumor (CPT).**

(A) Left panel: Principal component analysis of pseudobulk RNA seq data from epithelial cells of disease-free choroid plexus (ChP, circles), choroid plexus papilloma (CPP, triangles) and carcinoma (CPC, squares) samples, colored by methylation profile (green = pedB, blue = adult, gold = disease-free ChP). Right panel: Clustered heatmap showing gene expression similarity among samples. Top rows indicate patient sample type and methylation profile. (B) Venn diagrams showing degree of overlap of upregulated (top diagram) or downregulated (bottom diagram) DEGs between pedB (salmon) and adult (blue) samples. (C, D) Volcano plots showing gene expression changes between adult (C) or pedB (D) and ChP. Red dots indicate differentially expressed genes (DEGs, adjusted *p* value < 0.05, log2 fold change >1, Wald test-DESeq2). (E, F) KEGG pathways upregulated in adult (E) or pedB (F). Y-axis indicates number of pathway genes among upregulated DEGS, x-axis indicates fold enrichment of pathway genes. Circle diameter indicates the percentage of pathway genes that are upregulated in sample type. Fill color indicates −log10 of false discovery rate (FDR). (G) Average expression of selected candidate MAP kinase and/or PI3 kinase pathway DEGs illustrated per sample in CPT samples and disease-free ChP. Source data are available online for this figure.

Distinct, but significantly overlapping sets of genes showed differential expression in adult and pedB tumors (Dataset EV1), with pedB epithelial cells showing more dysregulated gene expression than adult profile epithelial cells (Fig. 4B–D). Of 3220 genes significantly upregulated (expression >2-fold ChP expression, adjusted *p* value < 0.05) in pedB, 406 (12.6%) were also upregulated in adult profile tumors (Fig. 4B). These commonly upregulated DEGs account for 55% of the 738 genes upregulated in adult profile tumors. Likewise, 235 of the 1825 genes significantly downregulated in our pedB samples (13%) or 412 genes downregulated in adult (57%) are shared between the two sample sets. The larger number of significant differentially expressed genes in the pedB dataset, as well as a greater tendency towards gene upregulation as opposed to downregulation, can be seen in volcano plots of expression change vs. *p*-value for each methylation profile (Fig. 4C,D).

As a first step to assaying the robustness of our gene expression dataset, we sought to independently test a subset of differentially expressed genes by performing RT-qPCR on three disease-free ChP samples and six CPT samples (Fig. EV3A). The methylation profiles of the CPT samples comprise one adult, one pedA, and four pedB samples, so for analysis we grouped these samples by histological diagnosis: 3 CPP and 3 CPC. While results from the small sample size did not reach significance, RT-qPCR showed clear trends towards differential expression for genes with comparable upregulation in both tumor profiles (*STIL*, *PROM1*), greater upregulation in the adult profile (*BAIAP3*, *WNT2B*) and greater upregulation in the pedB profile (*WNT5B*, *TACR3*) (Fig. EV3B). In order to assess expression of these DEGs in a larger group of CPT samples, we obtained normalized counts from a previously published bulk sequenced CPT dataset (Thomas et al, 2021), with 3 control samples, 27 adult samples, and 12 pedB samples. Gene expression of *PROM1* was significantly upregulated in both CPT profile tumors relative to disease-free ChP, while *STIL* expression was only upregulated in pedB tumors compared to disease-free ChP. *BAIAP3* and *WNT2B* were preferentially upregulated in adult profile tumors, while *WNT5B* and *TACR3* were preferentially upregulated in pedB profile relative to disease-free ChP (Fig. EV3C).

We then expanded our analysis to the 20 differentially expressed coding genes with the smallest adjusted p values in each comparison (adult vs. ChP and pedB vs. ChP). This set of top significant DEGs comprised 34 genes (14 from the adult dataset, 14 from the pedB dataset, and 6 shared significant DEGs). Plotting average expression per sample of these genes showed that these gene expression changes were uniform across samples of a given methylation classification (Fig. EV4A,B). To then assess how well gene expression differences observed in our dataset predict gene

expression in a larger cohort of tumor samples, we assayed expression of this set of significant DEGs in the bulk sequenced tumor cohort (Fig. EV4C–E). We observed significant (adjusted *p* < 0.05) changes in expression for 65% (13 out of 20) of adult top significant DEGs in bulk sequenced adult profile tumors. For pedB profile tumors, 90% (18 of 20) top significant DEGs identified in our dataset were significantly differentially expressed in the bulk-seq data. Thus, snRNA-seq of CPTs, identifies methylation profile specific changes in gene expression.

Although driver mutations have not been identified for CPT to date, several genes have been implicated, including Notch receptors, *MYC*, *TP53*, *RAD54L*, *TAF12*, *NFYC*, and *PTEN* (Merve et al, 2019; Shannon et al, 2018; Tong et al, 2015). Surveying expression changes for these genes in our dataset revealed significant upregulation of *NOTCH1*, *NOTCH2*, *NOTCH3*, and *RAD54L* in pedB tumors relative to disease-free ChP (Appendix Table S2).

We next sought to gain insight into the molecular pathology of CPT, by identifying pathways and processes that might be altered in adult and/or pedB profile tumors. We performed KEGG pathway enrichment analysis on significantly upregulated genes in each tumor type. Not surprisingly, we observed more enriched pathways in pedB than adult (Fig. 4E,F; Appendix Table S3). Significantly enriched (FDR < 0.05, fold enrichment ≥2) pathways shared by adult and pedB include Neuroactive ligand-receptor interaction, ECM-receptor interaction, PI3K-Akt signaling pathway and Nicotine addiction. In both adult and pedB data sets the Nicotine addiction pathway showed the greatest fold enrichment. Inspection of the DEGs involved in this pathway revealed GABA receptor subunits, which are known to be upregulated in several cancers (Bhattacharya et al, 2021; Huang et al, 2023; Huang and Cao, 2022; Yang et al, 2023), as well as channel subunits for regulated $Ca2^+$, $Na^+$, and $K^+$ entry into the cell (Appendix Table S4). The ECM-receptor interaction pathway was also enriched in both tumor types. Given the importance of tumor-ECM interactions in cancer growth and metastasis, we investigated the identities and expression profiles of genes within this pathway. Adult and pedB samples appear to upregulate overlapping but distinct sets of these genes (Appendix Fig. S1A,B), which comprise numerous ECM components including collagens, laminins, agrin, integrin subunits, and proteoglycans.

Given the importance of signaling through the MAP kinase and PI3 kinase-AKT pathways and their potential as viable drug targets (Cohen et al, 2021; Glaviano et al, 2023; Vanhaesebroeck et al, 2021), we investigated the expression profiles of genes from these pathways across our sample set (Fig. 4G). All pedB samples showed upregulation of MAPK pathway genes, with samples CC5, CC4,

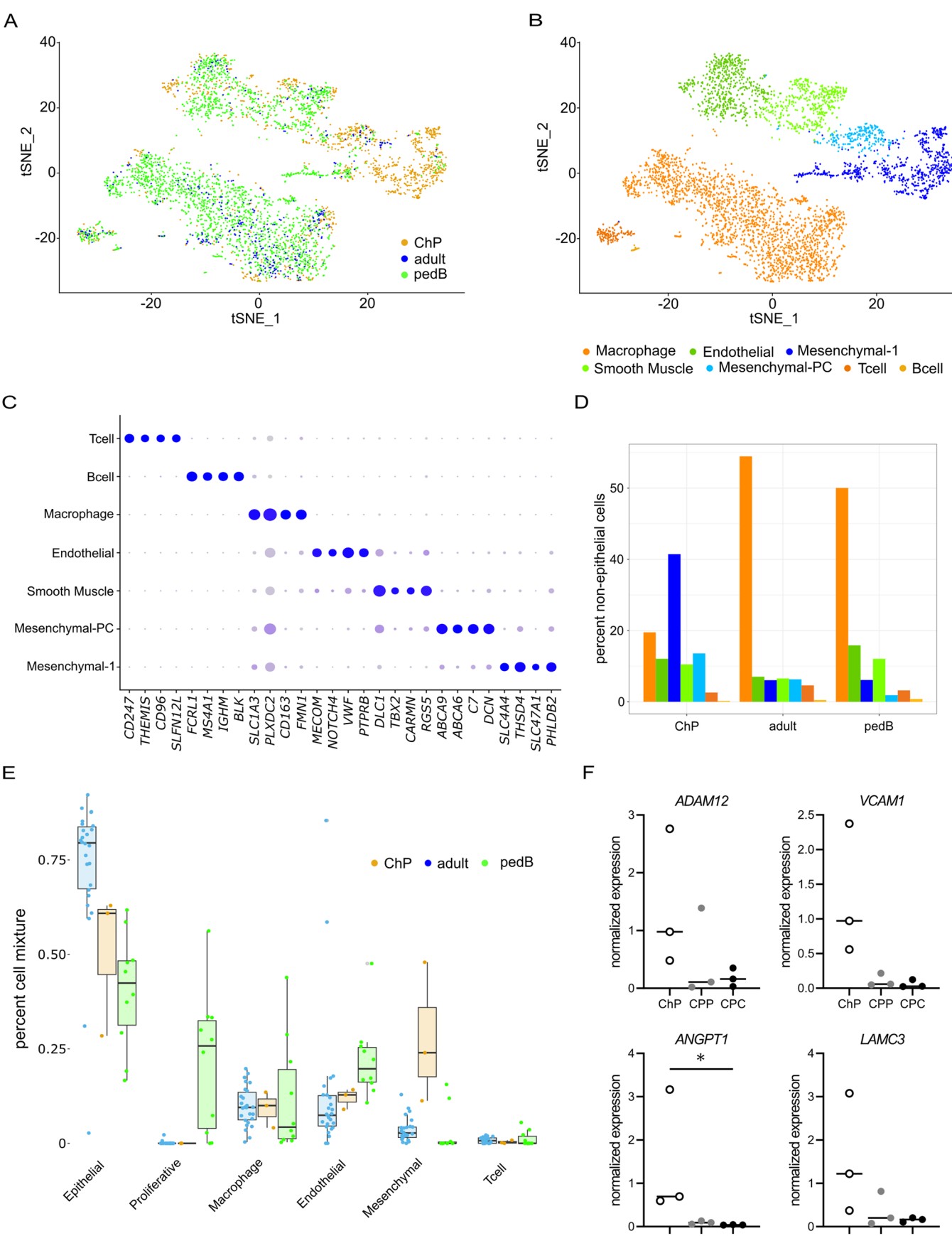

**Figure 5. Altered cellular tumor microenvironment in choroid plexus tumor.**

(A) Non-epithelial cells from disease-free choroid plexus (ChP, gold), adult profile (adult, blue) and pedB profile (pedB, green) tumor samples are differentially distributed on a tSNE plot. (B, C) Non-epithelial cell types (B) were identified by their expression of lowest adjusted *p* value cluster specific markers (C). (D) Column chart presenting cell type composition of ChP (left), adult (middle) and pedB (right) tumor microenvironment. (E) Deconvolved cell mixtures of bulk sequenced samples. Estimated cell prevalences of ChP cell types from bulk sequenced disease-free ChP (gold, $n = 3$ biological replicates), adult (blue, $n = 27$ biological replicates) and pedB (green, $n = 12$ biological replicates) samples. Boxplots indicate 25th through 75th percentile, bold black line indicates median. Whiskers show range of data. (F) Normalized mRNA expression levels of selected mesenchymal candidate genes in disease-free ChP, choroid plexus papilloma (CPP) and carcinoma (CPC) samples, as measured by RT-qPCR. $n = 3$ biological replicates (3 technical replicates per sample). Bars represent median; Kruskal–Wallis test, *$p \leq 0.05$ (*ANGPT1* ChP vs. CPC $p = 0.0219$). Source data are available online for this figure.

and CC2 showing the largest number of upregulated genes. PI3 kinase-AKT pathway genes were overexpressed in all of the tumor samples, with pedB samples showing more upregulated genes.

We next profiled DEGs involved in the best-known function of the ChP, production of CSF. CSF production requires (i) an impermeable barrier separating the CSF from the stroma and (ii) solute transporters to bring about the directional transport of water and solutes across that barrier (Ho et al, 2012). A targeted analysis of 7 genes regulating epithelial cell tight junctions (Furuse, 2010) showed significant downregulation of 4 tight junction components, *CLDN5*, *CLDN2*, *CLDN1*, and *OCLN* in pedB epithelial cells (Appendix Table S5). In addition, of the 1235 significant DEGs in adult profile epithelial cells, 29 were solute carrier family (SLC) transporter genes, while 122 of the 5045 pedB significant DEGs were SLC genes (Fig. 4B, Dataset EV1). Many of the lowest *p*-value transport proteins upregulated in CPT suggest changes in cell metabolism (glutamate and other amino acids, sugar, lactate and ADP/ATP transporters) as well as pH and ion balance (bicarbonate, sulfate, $Na^+$, $Ca^{2+}$, $H^+$, and $Cl^-$ transporters (Appendix Table S6).

## Altered cell states in choroid plexus tumor microenvironment

Along with the ECM, non-cancerous cells, including endothelial, mesenchymal and immune cells, form a tumor microenvironment (TME) that can influence the growth, survival, and metastasis of cancer cells (Jin and Jin, 2020). In order to probe the cellular and transcriptomic makeup of the TME in CPT, we repeated the clustering and dimensionality reduction steps on our dataset after removing epithelial cells. Non-epithelial cells form six clusters, identified by marker expression as mesenchymal-1, mesenchymal-PC, smooth muscle, endothelial, macrophage and T cells (Fig. 5A–C). Quantifying the cells in these clusters by sample type, we observed an apparent expansion of macrophage (adult and pedB) as well as endothelial and smooth muscle cells (pedB only) in CPT samples, at the expense of mesenchymal cells (Fig. 5D). Because of the small sample size, we sought to confirm these changes in CPT cellular composition by deconvolving the previously published bulk sequencing data and estimating cell fractions (Newman et al, 2019). While we did not observe consistent changes in CPT macrophage cell numbers, the deconvolved bulk sequencing data show a clear decrease in average mesenchymal cell signatures in CPT (ChP 27.7%, adult 3.6%, pedB 2.8%, adjusted *p*-value ChP vs. pedB = 0.002), as well as the marked increase of proliferative (ChP 0% vs pedB 21.5%, adjusted *p*-value = 0.006) and endothelial cells (ChP 12.7% vs pedB 22.1%, adjusted *p*-value = 0.0012) in pedB samples (Fig. 5E). We next

assessed the mRNA expression of selected mesenchymal cell markers by RT-qPCR. Normalized expression for *ADAM12*, *VCAM1*, *ANGPT1*, and *LAMC3* showed clear reduction trends in CPP and CPC samples compared to disease-free ChP. The only comparison that reached significance was *ANGPT1* expression in ChP compared to CPC (Fig. 5F).

We performed differential gene expression analysis for each of the major non-epithelial ChP cell types. Unsurprisingly, we detect fewer significant DEGs in endothelial, mesenchymal, and macrophage cells from CPT than are detected in CPT epithelial cells (Appendix Table S7; Fig. 4B). For each cell type, more significant DEGs were identified in pedB profile cells than adult profile cells (Appendix Table S7).

For many tumors, tumor-associated macrophages (TAMs) constitute a major component of the TME and influence tumor growth, metastasis, and immune surveillance (Zhukova et al, 2020). In order to identify macrophage cell state changes in CPT, we next characterized gene expression in macrophages from ChP, adult, and pedB samples. We observed decreased expression of many genes involved in inflammatory responses, including MHCII genes and their regulator *CITTA* along with *CD74* and complement cascade genes (*C1QA, C1QB, C1QC*) in the pedB cohort (Fig. 6A). Using the same CPT cohort used for mesenchymal markers, we quantified the mRNA expression levels of *C1QA*, *C1QB*, *HLA-DRA*, and *HLA-DPB1*. All tested inflammation-related markers showed decreased expression in CPC samples, but these differences were only significant for *C1QA* and *C1QB* (Fig. 6B). A similar reduction trend of the same markers was observed when we compared bulk sequenced pedB samples to disease-free ChP, though these differences were not statistically significant (Fig. 6C). It is well accepted that many tumors can produce factors that recruit TAMs and push them towards an anti-inflammatory/wound healing phenotype (Argyle and Kitamura, 2018). It is therefore worth noting that anti-inflammatory chemokines including *CSF1*, *CD276*, *VEGFA*, and *CX3CL1* were upregulated in pedB tumor epithelial cells (Table 2). All together these findings suggest that pedB macrophages could be less competent to mediate an inflammatory response compared to adult tumors.

In order to further investigate this possibility, we performed immunohistochemistry for the macrophage marker IBA1 on sections of paraffin preserved human CPP and CPC (Appendix Table S8) and examined them by transmitted light microscopy. Due to the limited availability of similarly prepared disease-free ChP sections, we focused our analysis on comparing CPP vs. CPC macrophage morphology. In CPP sections, we observed compact and round or oval-shaped IBA1-positive (IBA1$^+$) cells (Fig. 6D, top row), suggesting a more pro-inflammatory/anti-tumor phenotype (Costa et al, 2023;

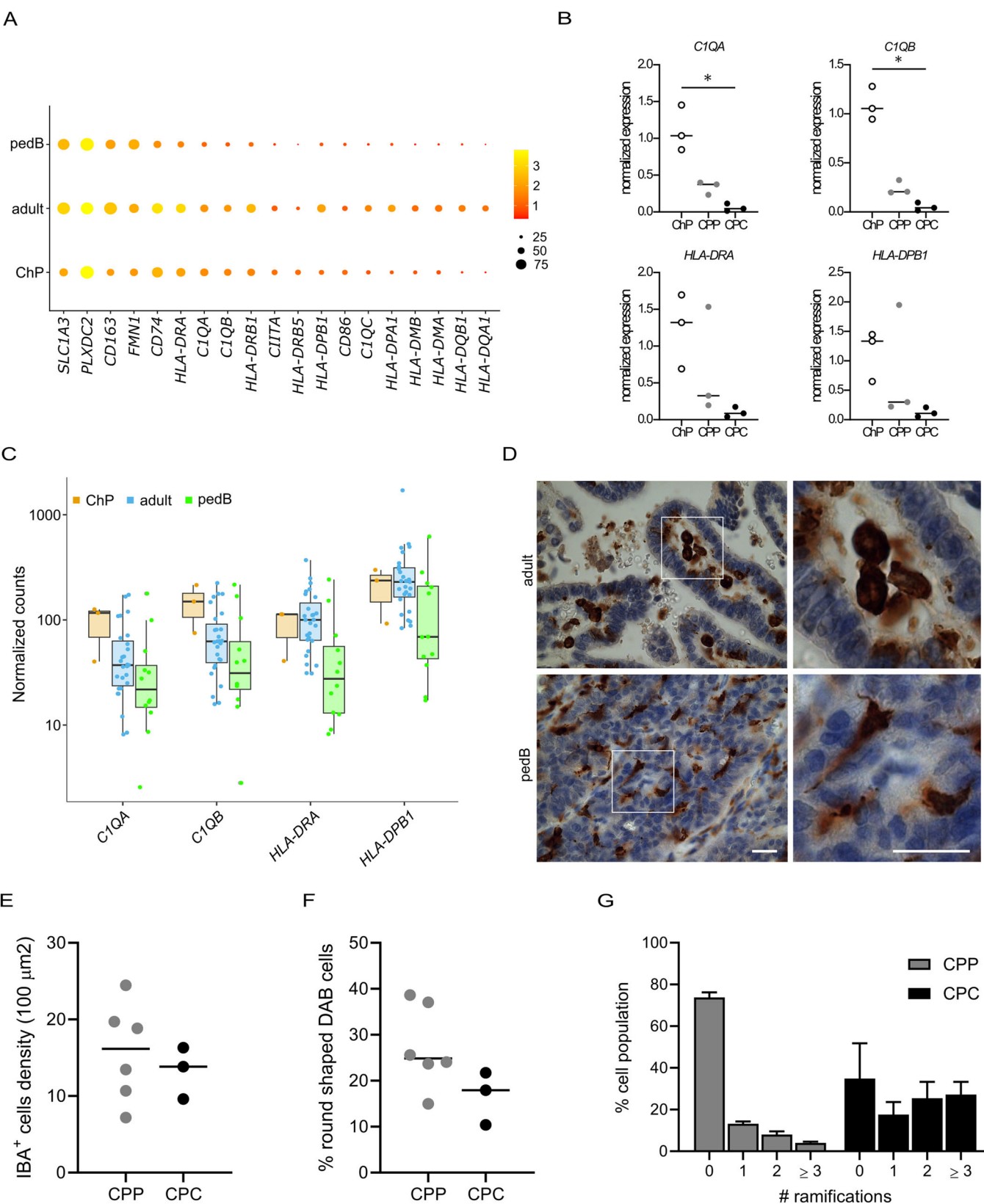

**Figure 6. Morphological and transcriptional changes in choroid plexus tumor macrophages suggest altered function.**

(A) Dotplot showing average expression and percent expressing cells of macrophage markers and genes involved in inflammatory responses in macrophages from disease-free choroid plexus (ChP), adult profile tumors (adult), or pedB profile tumors (pedB). (B) Normalized mRNA expression levels of complement (*C1QA, C1QB*) and MCHII (*HLA-DRA, HLA-DRB1*) in disease-free ChP, choroid plexus papilloma (CPP) and carcinoma (CPC) samples, as measured by RT-qPCR. $n = 3$ biological replicates (3 technical replicates per sample). Bars represent median. Kruskal–Wallis test adjusted p-values, *$p \leq 0.05$ (*C1QA* ChP vs. CPC $p = 0.02$, *C1QB* ChP vs. CPC $p = 0.02$). (C) Normalized counts data for *C1QA, C1QB, HLA-DRA,* and *HLA-DRB1* for bulk sequenced ChP ($n = 3$ biological replicates), adult ($n = 27$ biological replicates), and pedB ($n = 12$ biological replicates) profile samples. Boxplots indicate 25th through 75th percentile, bold black line indicates median. Whiskers show range of data. (D) Representative photomicrographs illustrating IBA1-positive cells in CPP (top panels) and CPC (bottom) tissue sections. Scale bar $= 25\,\mu m$. (E–G) Quantification of total IBA1$^+$ cell density (E) and percent round cells (F). Percentage of IB1-positive ramifications (0, no ramifications to 3 ramifications per cell) (G). Gray filled dots and bars $=$ CPP, black filled dots and bars $=$ CPC. $n = 3$-6 biological replicates (1 whole tumor section per sample). Bars represent median (E, F); Bars represent mean $\pm$ SEM (G). Source data are available online for this figure.

Donadon et al, 2020; McWhorter et al, 2013; Rostam et al, 2017; Vereyken et al, 2011). These cells were predominantly located in the stroma or at the stromal surface of the CPP epithelial cells (Fig. 6D, top row). By contrast, in CPC sections many elongated or irregularly shaped IBA1$^+$ cells were distributed throughout the tissue, where a subset seemed to extend several processes (Fig. 6D, bottom row). This morphology suggests a wound healing/anti-inflammatory bias (Costa et al, 2023; Donadon et al, 2020; McWhorter et al, 2013; Rostam et al, 2017; Vereyken et al, 2011). We next used an automated slide scanning microscope and manual image analysis to quantitate macrophage presence and morphology in CPT sections. While we did not find significant differences in either total macrophage density or the percent of round macrophages (Fig. 6E,F), we observed significantly increased ramification of CPC IBA1$^+$ macrophages when compared to CPP ($p$-value $< 2.2e-16$; Fig. 6G). Taken together, these data suggest that pedB macrophages exhibit a wound healing/anti-inflammatory bias.

While macrophages are the largest cellular component of the CPT microenvironment, it also includes substantial numbers of endothelial and mesenchymal cells. In order to get an overview of the complex signaling occurring among CPT cells we used CellphoneDB (Efremova et al, 2020) to infer potential ligand-receptor interactions among ChP, adult, and pedB profile cell types. To get an overview of the patterns of intercellular communication within CPT, we first deployed the statistical method within the CellphoneDB package, which uses a permutation approach to identify relevant interactions. Visual inspection of circos diagrams depicting interactions within each sample type suggests similar patterns of cell–cell interactions among ChP, adult, and pedB profile samples, with the addition of interactions with the proliferative cells in the adult and pedB plots, and the observation that pedB proliferative cells appear to take part in more interactions than adult proliferative cells (Fig. 7A–C). However, dotplots showing the top 20 interactions identified in each sample type reveal diverged interaction sets (Fig. 7D–F). The top 20 interactions in disease-free ChP are primarily driven by epithelial cells and feature secreted signaling molecules (PGD2, Glutamate, Cholesterol, FGFs, WNT5A) and membrane bound signaling and adhesion molecules (TENM2, NLGN1, Ephrins). Despite being less represented within the tumor types (Fig. 4), mesenchymal cells showed many interactions within each tumor type. Macrophage and epithelial cells seemed to also take part in more interactions in adult types, especially when compared to pedB types. Interestingly, the main tumor interaction networks include ECM components previously identified in the KEGG analysis (SPP1, FN1, collagens), cell–cell interactions and cholesterol along with apolipoprotein (adult).

We next used the DEG module of CellphoneDB to focus on specific interactions involving differentially expressed genes in CPT epithelial lineage cells. Specifically, we looked at interactions in disease-free ChP that involved genes with significantly reduced expression in adult or pedB tumor epithelial cells (Fig. EV5, top row) and interactions in each tumor type involving genes with significantly increased expression in tumor epithelial cells (Fig. EV5, bottom row). This approach allows the identification of interactions that are reduced or absent in CPT as well as upregulated or novel interactions. The most significant ChP interactions involving genes downregulated in adult profile epithelial cells include NGLN-NRXN, SLIT-ROBO, PGF-FN1, and cholesterol signaling (Fig. EV5, upper left). Genes downregulated in pedB profile epithelial cells take part in IGF, ERB-B, ephrin, and WNT signaling interactions in disease-free ChP (Fig. EV5, upper right). Turning to tumor interactions that involve upregulated DEGs, in adult profile tumors epithelial cell DEGs participate in interactions involving ECM components, cell adhesion, ERB-B signaling and glutamate transport (Fig. EV5, lower left). In pedB profile tumors, epithelial DEGs participating in significant interactions include CLSTN2, VEGFA, FGF1, collagens, WNT5B, and NRXN3 (Fig. EV5, lower right).

# Discussion

## A cellular atlas of human choroid plexus tumors

This atlas presents the major cell types in normal and cancerous ChP, which include epithelial, vascular, mesenchymal and immune cell types. Prior to batch correction, we observed considerable epithelial cell heterogeneity. Among CPT samples, we observed considerable inter-individual heterogeneity, but epithelial cells also broadly group by tumor type and methylation classification profile. Presumably, some of the heterogeneity observed reflects uncontrolled external variables including the elapsed time between sample excision and cryopreservation, particularly considering the small number of samples used in this study. However, the fact that epithelial cells largely group by tumor type while each non-epithelial cell type forms a single cluster is consistent with the notion that the observed heterogeneity at least partially reflects intrinsic differences between individuals and among tumor types.

Batch correction, integration and clustering each group of samples revealed additional cellular diversity. Disease-free ChP samples form 3 clusters that express mesenchymal markers, showing a previously undescribed heterogeneity of this cell type in human ChP

**Table 2. Tumor-derived macrophage inhibitor expression in choroid plexus tumor epithelial cells.**

| Gene | L2FC (adult_Epithelial) | padj (adult_Epithelial) | L2FC (pedB_Epithelial) | padj (pedB_Epithelial) |
|------|------------------------|-------------------------|------------------------|------------------------|
| CSF1 | 0.46 | 0.44 | 1.29 | 0.05 |
| VEGFA | −0.43 | 0.47 | 1.25 | 0.01 |
| CX3CL1 | 0.19 | 0.79 | 1.06 | 0.04 |
| CD276 | −0.11 | 0.87 | 1.92 | 0.01 |

Fold expression changes for macrophage anti-inflammatory factor genes in adult profile (adult) and pedB profile (pedB) epithelial cells. *L2FC* log2 fold change, *padj* adjusted *p*-value.

(Dani et al, 2021; Yang et al, 2021). Mesenchymal cells are extremely divergent and exhibit organ-specific patterns of gene expression (Baek et al, 2022; Bruijn et al, 2020), which prevented a positive identification of these mesenchymal cell types either manually or using automated methods. One cluster expressed pericyte markers including angiopoietin, *RGS5*, and *PDGFRB*, while the other two express distinct subsets of fibroblast-like cell markers. In addition, two clusters in integrated disease-free ChP samples express endothelial cell markers including *PECAM1*, *VWF*, and *FLT1*. One of these clusters contains cells expressing markers of smooth muscle or other arterial cells including *ACTA2*, *MYH11*, *DKK2*, and *IGFBP3*. The second expresses markers of fenestrated capillaries (*PLVAP*) and venous cell markers (*TSHZ2*, *VWA1*) (Hennigs et al, 2021; Schupp et al, 2021). Altogether, the identified cell sub-type markers will facilitate the future morphological and functional analysis of these cells and they will be useful in studies of the induction and recruitment of non-epithelial cells in the developing ChP. Integrating all of the libraries reveals a tumor-specific, epithelial-lineage, proliferative cell in adult and pedB profile tumors, as well as reduced presence of mesenchymal cells in tumor cells.

## A transcriptional atlas of choroid plexus tumors

This atlas also provides an overview and comparison of ChP, adult and pedB inferred CNVs, transcriptomes, DEGs and pathway enrichment. The most common pedB profile CNVs inferred from our dataset include gain of 1p, 1q, 4p, 4q, 12p, 12q and 20p, each of which was frequently observed for pedB profile samples in a larger bulk sequenced dataset (Thomas et al, 2021). For adult profile tumors, we observe frequent gain of chromosome arms 9p and 9q, along with loss of 21p and 21q, also consistent with the earlier bulk sequencing dataset, with the exception of 21q loss, which was not reported previously (Thomas et al, 2021).

Gene expression in pedB samples is more divergent from normal ChP than that of adult samples. Enrichment analysis of CPT epithelial cell DEGs suggest that pathways involved in cell growth and proliferation (PI3K-Akt signaling signaling, Rap1 signaling), ion homeostasis (Nicotine addition, Glutamatergic synapse) and ECM interactions are upregulated in adult profile CPT. DEGs from pedB samples showed enrichment for pathways including NOTCH, MAPK, PI3K-AKT, RAS, RAP1, and Calcium signaling, pathways involving ion movement and downstream consequences (Nicotine addiction, Morphine addiction, GABAergic synapse), as well as pathways involving local interactions (ECM-receptor interaction, Focal adhesion).

A promising line of research might be to investigate the consequences of altering signaling through enriched pathways, including PI3K-AKT signaling, MAPK signaling, and RAS signaling, in ChP tumor cells. For instance, most of the samples in our dataset

showed some level of PI3K-AKT signaling activation, with the highest and broadest activation being observed in pedB samples. This pathway is altered in ~50% of cancers, where it drives tumor development, progression, and resistance to therapy (Glaviano et al, 2023). Further study of this pathway in CPT seems warranted.

Both adult and pedB tumors show enrichment for genes in the ECM-receptor interaction pathway. ECM components have been shown to have profound effects on the survival, proliferation, morphology and migration of normal and cancerous cells (Huang et al, 2021; Zhao et al, 2021). It is therefore interesting that ChP, adult profile and pedB profile epithelial cells generate distinct ECMs.

The best characterized function of the ChP is the production of CSF, a process dependent upon the unidirectional transport of $Na^+$, $Cl^-$ and $HCO_3^-$ ions, which creates an osmotic gradient that drives the movement of water molecules. Notably, several transporters for these ions are upregulated in CPTs, including *SLC4A7*, *SLC4A8*, and *SLC8A1*. Both CPP and CPC are often accompanied by hydrocephalus (Konar et al, 2021), presumably through some combination of CSF overproduction, physical blockage of drainage routes by the tumor mass, and CSF malabsorption. Because the resulting increase in intracranial pressure can cause permanent brain damage, it is often a more urgent cause for surgical intervention than the growing tumor itself. Inhibition of $Na^+$ transport (Davson and Segal, 1970; Pollay et al, 1985), or knock out of the $HCO_3^-$ transporters *Slc4A5* or *Slc4A10* (Carroll et al, 2022; Kao et al, 2011) have each been shown to reduce CSF secretion. Additional research is required to determine whether inhibitors of $Na^+$, $Cl^-$ and $HCO_3^-$ ion transporters can reduce hydrocephalus in CPT patients. This might allow physicians to delay surgery on newborn CPT patients, potentially improving outcomes. Recently, a number of labs have begun targeting bicarbonate transporters because they are upregulated in multiple cancer types, where they promote tumor cell growth, survival, and immunosuppression (Cappellesso et al, 2022; McIntyre et al, 2016; Pollay et al, 1985). Inhibitors of $HCO_3^-$ ion transporters might therefore have multiple benefits in the treatment of CPT.

Several transporters of glutamate, including *SLC1A7*, *SLC7A1*, *SLC7A11*, and *SLC1A3* are upregulated in pedB tumors. Inhibitors of these proteins also merit investigation, given the importance of glutamate in a wide variety of tumors (Fantone et al, 2024; Stepulak et al, 2014; Xu et al, 2013).

## Characterizing the tumor microenvironment of CPT

CPTs with the pedB methylation profile overexpress macrophage chemoattractants, including *CSF1*, *VEGFA*, and *CD276* that have been shown to push macrophages toward an anti-inflammatory

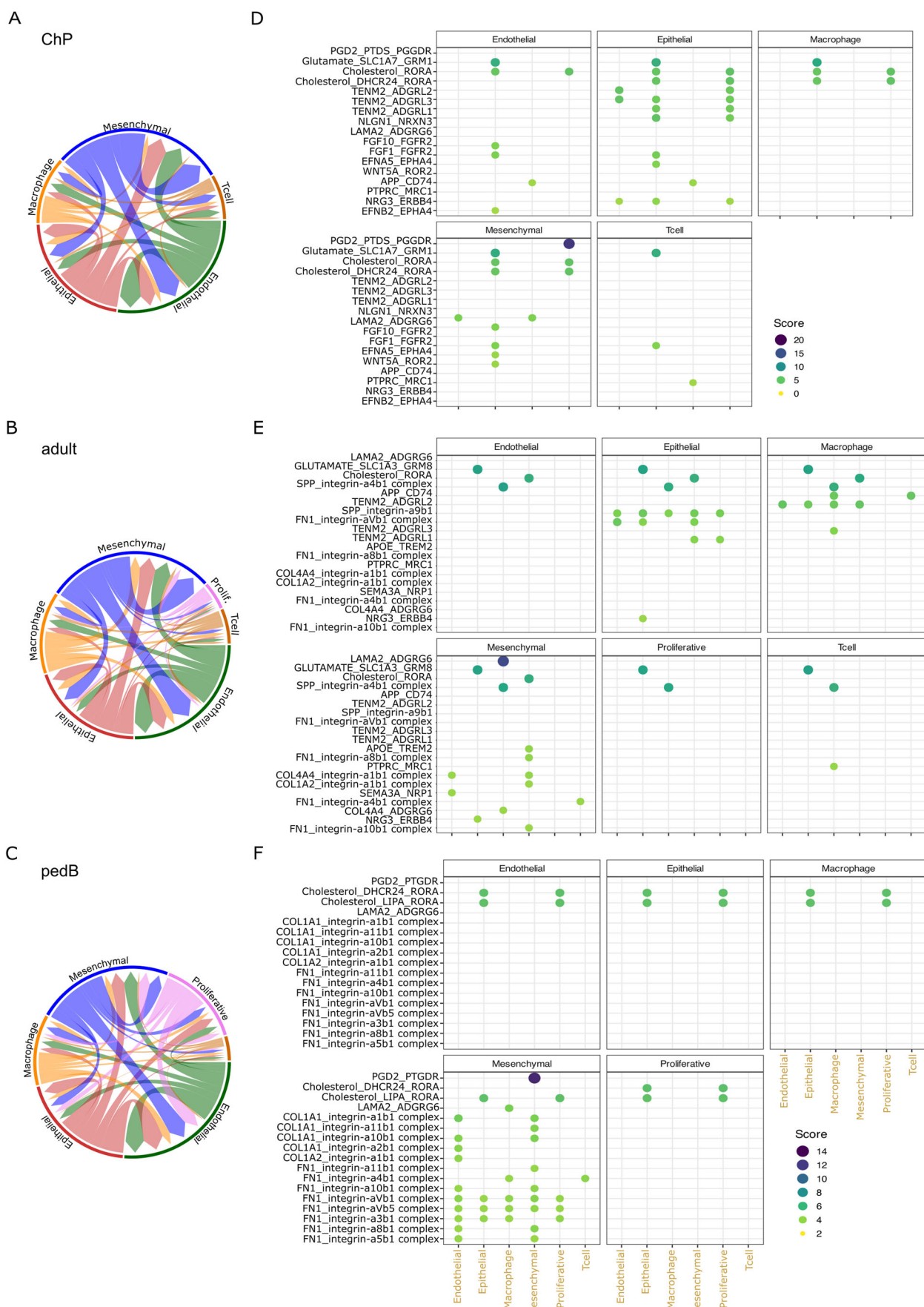

**Figure 7. Altered patterns of cell–cell signaling in choroid plexus tumor (CPT).**

(A–C) Circos diagrams showing inferred cell–cell interactions in disease-free choroid plexus (ChP, **A**), adult profile choroid plexus tumor (adult, **B**), and pedB profile tumors (pedB, **C**). Arrow color = cell type of origin, thickness = number of ligand-receptor pairs. (**D–F**) Plot of inferred interactions as visualized by CellSignalR package for disease-free ChP (**D**), adult (**E**), and pedB (**F**) samples, grouped by source cell type (black lettering) and sub-divided by target cell types (gold lettering).

phenotype (Durlanik et al, 2021; Wang et al, 2021), suggesting one possible mechanism by which pedB tumors evade immune surveillance. Consistent with this notion, pedB macrophages fail to express markers of immunocompetency, and do not assume the morphology of activated cells. Attempting to slow either the growth or recurrence of CPC by targeting the recruitment and polarization of pedB TAMs with (for instance) VEGFA inhibitors could be an interesting aim for future research.

Finally, we observe altered cell–cell interactions among CPT cells, that involve tumor-specific ECM components, as well as developmentally relevant signaling molecules including NOTCH ligands, BMPs, FGFs and semaphorins.

Taken together, this atlas serves as a reference for researchers studying normal or diseased choroid plexus, giving an overview of cellular heterogeneity in human disease-free ChP, and CPT. Future studies will be needed to assess the spatial distribution of the relevant cell types and proteins in CPT, and to demonstrate the functional significance of the pathways identified herein. This future work would benefit from new standard surgical operating procedures and tissue banks to make high-quality, well-preserved samples from these rare but potentially devastating tumors more widely available to researchers.

## Methods

### Human materials

De-identified CPTs were obtained from the archive of the Institute of Neuropathology in Münster (Münster, Germany) and from the Nationales Centrum für Tumorerkrankungen (NCT) (Heidelberg, Germany). Histopathology was reviewed according to current WHO criteria. Disease-free ChPs were kindly provided by The Netherlands Brain Bank (Amsterdam, the Netherlands). The handling of all tissues and experiments were complied with ethical guidelines and regulations for the research use of human tissue issued by the Medical Faculty of the Heidelberg University (S-753/2019) and conform to the WMA Declaration of Helsinki and the Belmont Report. Information about all tissue specimens is provided in Table 1.

### Droplet-based single-nucleus sequencing

Frozen tissue was thawed in ice-cold wash buffer (0.32 M sucrose, 5 mM $CaCl_2$, 3 mM magnesium acetate, 2.0 mM EGTA, 10 mM Tris-HCl pH 8.0) + 0.001% Triton-X and 1 mM DTT. Thawed tissue fragments were transferred to a glass dounce on ice and given 10 strokes each with an A-type pestle and a B-type pestle. The resulting nuclei suspension was sequentially filtered through 100 and 40 μm filters and pelleted by centrifugation for 5 min at $555 \times g$ and 4 °C. Pellets were washed by resuspension in wash buffer and centrifugation 2–4 times. After the final wash, pellets were resuspended in sterile filtered nuclei storage buffer (0.43 M sucrose, 70 mM KCl, 2 mM $MgCl_2$, 10 mM Tris-HCl pH 7.2, 5 mM EGTA).

Single nucleus expression libraries were made from 5 to 10 K nuclei/sample using the Chromium Single Cell 3' Library & Gel Bead kit v3.1 (10x Genomics) according to the manufacturer's protocol. Briefly, cells were partitioned into Gel Beads in Emulsion in the Chromium instrument for lysis and barcoding. Amplification, fragmentation, adapter ligation and index library PCR of single nucleus cDNA was performed in a 96-well thermocycler with heated lid. Library size and quantity were measured by Tapestation (Agilent) and Qubit Fluorometric quantification (ThermoFisher), respectively. Libraries were pooled and sequenced using a NovaSeq 6000 (Illumina).

### Sequence analysis

Sequencing reads were aligned and expression matrices were generated using CellRanger 6.0. Cells were filtered for quality (650–20,000 UMI/cell, 800–5000 detected genes/cell, percent mitochondrial genes <5, Appendix Table S1). Cell clustering and dimensionality reduction were performed using Seurat 4.3. Copy number profiling was performed on the merged raw gene expression counts matrix from single nucleus data of CPT samples ($n = 11$) using InferCNV of the Trinity CTAT Project (https://github.com/broadinstitute/inferCNV). To improve the visualization and decrease sparsity, groups of 5 tumor cells were merged into meta-cells as described previously (Blanco-Carmona et al, 2023). All cells from disease-free ChP samples ($n = 4$) were used as reference control.

Single-cell CNV profiles were compared to bulk derived from methylation data based on visual inspection.

The verified amplifications and gains across chromosome fragments were collected for each case.

Dataset integration was performed according to Stuart et al (Stuart et al, 2019) using tools built into Seurat. Pseudobulk differentially expressed genes (adjusted $p$ value < 0.05, log2 fold change >1) were identified using DESeq2. KEGG enrichment analysis was performed using ShinyGO 0.77 (FDR = 0.05, background gene set = all genes expressed in the 2 comparison populations with baseMean >1.5). Cell–cell interaction networks were identified using CellphoneDB (Efremova et al, 2020).

Data analysis was not performed blind from sample conditions.

### RT-qPCR

RT-qPCR was performed following the detailed procedure as previously described (Ho and Patrizi, 2021). Primer sequences are below:

Target gene primer sequences are from PrimerBank (Wang et al, 2012). Reference gene primer sequences and their sources were described previously (Ho and Patrizi, 2021).

| Target | Primer1 | Primer2 | ID |
|--------|---------|---------|-----|
| *ADAM12* | AACCTCGCTGCAAAGAATGTG | CTCTGAAACTCTCGGTTGTCTG | 194733767c3 |
| *VCAM1* | CAGTAAGGCAGGCTGTAAAAGA | TGGAGCTGGTAGACCCTCG | 315434270c3 |
| *ANGPT1* | TCGTGAGAGTACGACAGACCA | TCTCCGACTTCATGTTTTCCAC | 21328452c2 |
| *LAMC3* | CCCACCTCGGTCAACATCAC | GAGGCGCTGTAGAACTGGTA | 110611155c1 |
| *C1QA* | TCTGCACTGTACCCGGCTA | CCCTGGTAAATGTGACCCTTTT | 87298824c1 |
| *C1QB* | TTCTGTGACTATGCCTACAACAC | GCCCAGTAGTGAGTTCTTGTC | 87298827c3 |
| *HLA-DRA* | ATACTCCGATCACCAATGTACCT | GACTGTCTCTGACACTCCTGT | 301171411c3 |
| *HLA-DPB1* | ATTCTGCCCGGAGTAAGACAT | TCGTTGAACTTTCTTGCTCCTC | 334688863c3 |

## Immunochemistry staining, microscopy, and analysis

3 μm sections of formalin-fixed paraffin-embedded (FFPE) human tissues (Appendix Table S8) were stained for rabbit anti-IBA1 (1:2000, ThermoFisher Scientific, #711504, RRID: AB_2734811) on the automated DAKO Autostainer Link 48 AS480. In brief, the staining procedure started with heat-induced epitope retrieval pretreatment using citric acid buffer (pH 6.0) at 80 ℃, then samples were incubated with primary antibody diluted in 1% BSA for X time. Slides were then rinsed and incubated in streptavidin-conjugated with horseradish peroxidase secondary antibody (1:500, horse anti-rabbit, Vector Labs) followed by nickel-enhanced diaminobenzidine/peroxidase reaction. Nuclei were counterstained with haematoxylin.

Whole labeled tumor sections were scanned on a Zeiss Axioscan 7 slide scanner at ×20 magnification and analyzed with Fiji (Bethesda, Maryland). All IBA1-positive cells and all nuclei-positive cells were manually counted on a whole tumor section. The total branch number per IBA1-positive cell was also manually quantified. All quantifications were performed blind to genotype.

## Statistical analyses

Statistical analyses were conducted using GraphPad Prism (version 9.3.1). Appropriate statistical tests were determined based on the distribution of data and sample size. Nonparametric tests were chosen (Mann–Whitney and Kruskal–Wallis). Data are presented as mean ± standard error of the mean (SEM) or as median. The specific statistical test used is indicated in each figure legend.

## Data availability

Raw and processed human choroid plexus and human choroid plexus tumor sequencing files are available at the Gene Expression Omnibus (GEO) database GSE264154. Code is available at https://github.com/corticaltone/CPT_Atlas.

The source data of this paper are collected in the following database record: biostudies:S-SCDT-10_1038-S44318-024-00283-2.

## Peer review information

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

## Acknowledgements

We thank the German Cancer Research Center (DKFZ) Omics IT and Data Management core facility for computational resources and data storage, the Single Cell Open Lab (DKFZ) for assistance in library preparation, the Light Microscopy core facility (DKFZ) for providing the cryostats and microscopes and the Genomics and Proteomics Core Facility (DKFZ) for providing the RT-qPCR machine Roche LightCycler480. We thank Jana Henry, Marleen Trapp, Jo Schanz and other Patrizi group members for the support and helpful discussions. This research was supported by the Chica and Heinz Schaller Foundation (AP).

## Author contributions

**Anthony D Hill**: Conceptualization; Data curation; Formal analysis; Validation; Visualization; Writing—original draft; Project administration; Writing—review and editing. **Konstantin Okonechnikov**: Formal analysis; Visualization; Writing—review and editing. **Marla K Herr**: Formal analysis; Writing—review and editing. **Christian Thomas**: Resources; Validation; Writing—review and editing. **Supat Thongjuea**: Writing—review and editing. **Martin Hasselblatt**: Resources; Writing—review and editing. **Annarita Patrizi**: Conceptualization; Resources; Formal analysis; Supervision; Funding acquisition; Validation; Project administration; Writing—review and editing.

Source data underlying figure panels in this paper may have individual authorship assigned. Where available, figure panel/source data authorship is listed in the following database record: biostudies:S-SCDT-10_1038-S44318-024-00283-2.

## Funding

## Disclosure and competing interests statement

The authors declare no competing interests.

# Expanded View Figures

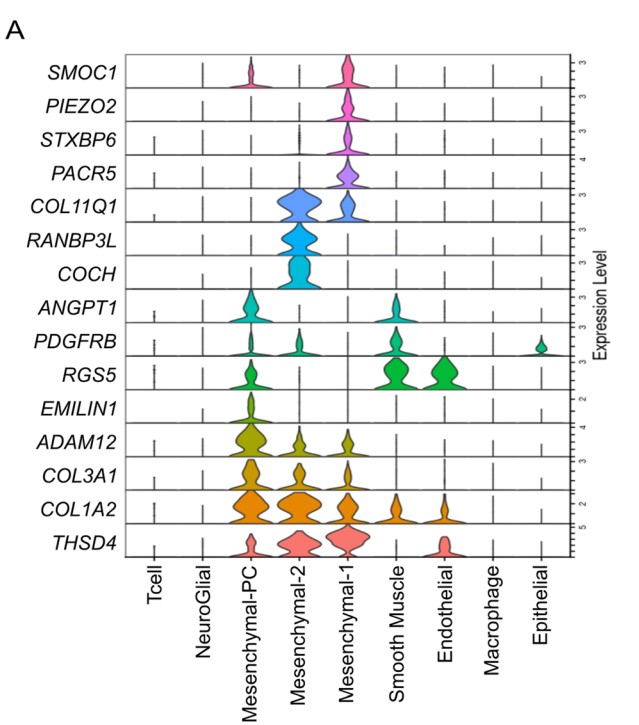

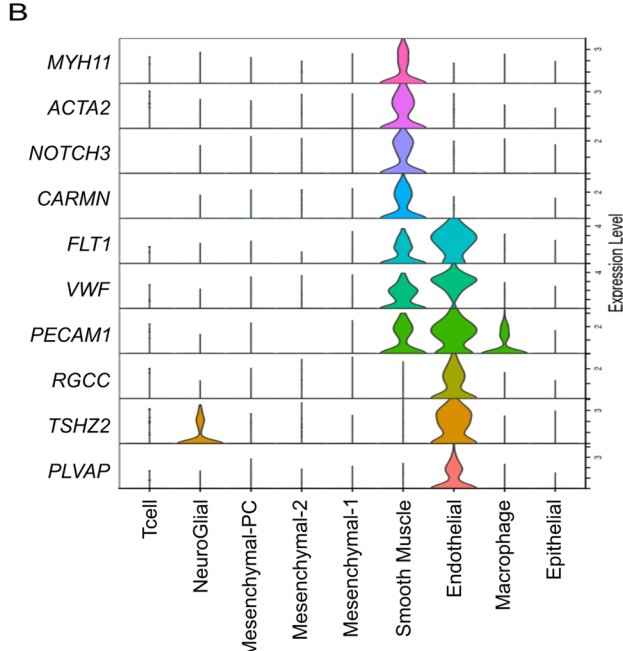

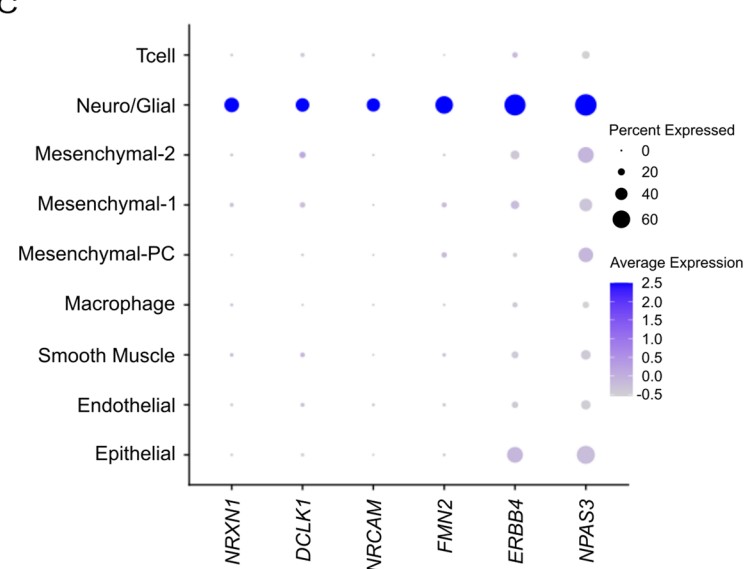

**Figure EV1. Cellular composition of disease-free human choroid plexus (ChP) and choroid plexus tumors.**

(A) Violin plots showing mesenchymal cell marker expression in disease-free human ChP. (B) Violin plots of endothelial and smooth muscle cell marker expression in disease-free human ChP. (C) Dotplot of neuronal and glial cell markers in human disease-free human ChP ($n = 4$ biological replicates).

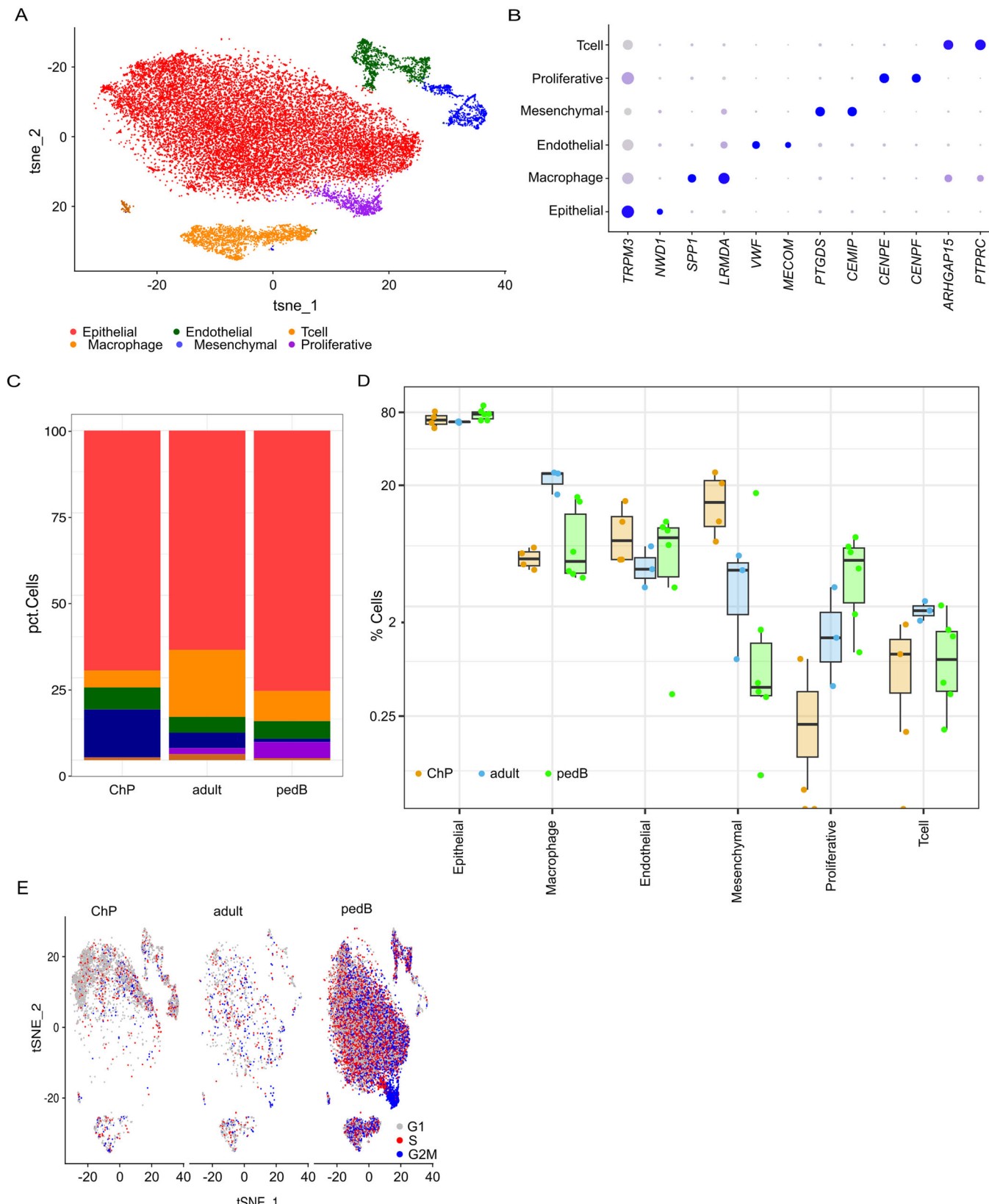

◀  **Figure EV2.  Cellular composition of choroid plexus tumors (CPTs).**

(A) Cell clustering and dimension reduction of the entire batch corrected dataset of disease-free choroid plexus (ChP) and CPTs. (B) Dotplot of low *p*-value cluster specific gene expression in each cell cluster (Wilcoxon rank sum test-Seurat). (C) Relative cell number of major ChP cell types in CPT samples with adult or pedB methylation profiles, as well as disease-free ChP. Boxplots indicate 25th through 75th percentile, bold black line indicates median. Whiskers show range of data. (D) Percent of major cell types identified in each disease-free human ChP (gold, $n = 3$ biological replicates) or CPT sample, grouped by methylation (adult profile, blue; pedB profile, green; $n = 27$ and 12 biological replicates, respectively). (E) Cell cycle scoring by methylation profile and cell type. tSNE projections of cells in G1 (gray), S (red), or G2M (blue) phase of the cell cycle for disease-free ChP (left panel), adult profile CPT (middle panel), or pedB profile CPT (right) samples. (gold, $n = 3$ biological replicates), adult (blue, $n = 27$ biological replicates) and pedB (green, $n = 12$ biological replicates).

**A**

| Gene | L2FC (adult_Epithelial) | padj (adult_Epithelial) | L2FC (pedB_Epithelial) | padj (pedB_Epithelial) |
|---|---|---|---|---|
| STIL | 1.47 | 6.17E-15 | 3.80 | 2.27E-13 |
| PROM1 | 3.01 | 0.04 | 4.82 | 5.10E-09 |
| BAIAP3 | 2.75 | 4.22E-03 | 0.46 | 0.50 |
| WNT2B | 3.22 | 8.36E-08 | 1.95 | 7.60E-05 |
| WNT5B | 7.09 | 6.17E-15 | 9.63 | 1.18E-35 |
| GNB3 | 6.19 | 1.15E-06 | 9.44 | 3.23E-19 |

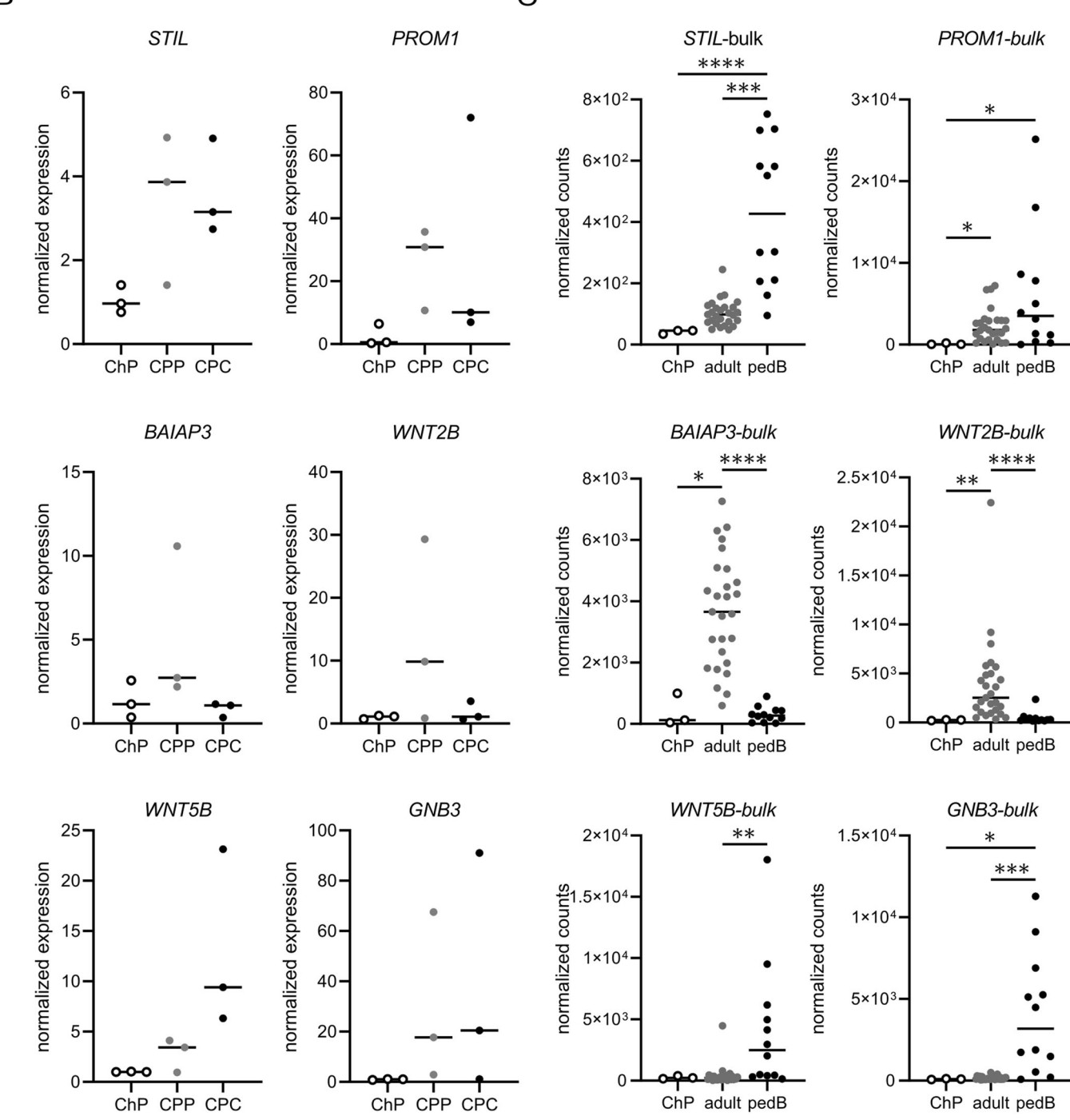

**Figure EV3. Expression of select differentially expressed genes identified in single nucleus (sn) analysis characterizing choroid plexus tumor (CPT) epithelial cells.**

(A) Expression fold change and adjusted *p*-values for selected genes in pseudo-bulk analyzed snRNAseq data for adult and pedB profile epithelial cells. L2FC = log2 fold (Wald test-DESeq2). (B, C) Relative mRNA expression levels of candidate genes preferentially expressed in CPT epithelial cells (top row), candidate genes preferentially represented in low-risk CPT epithelial cells (middle row) and candidate genes preferentially expressed in high-risk CPT epithelial cells (bottom row). Expression analysis was performed by RT-qPCR (B) or by normalizing counts from published bulk sequencing (C). ChP, disease-free choroid plexus; CPP, papilloma; CPC, carcinoma; $n = 3$ biological replicates (3 technical replicates per sample) (A); see Thomas et al, 2021 for data information (B). Bars represent median. Kruskal–Wallis test adjusted *p*-values: *$p \le 0.05$, **$p \le 0.01$, ***$p \le 0.001$, ****$p \le 0.0001$ (*STIL1*-bulk ChP vs. pedB $p < 0.0001$, adult vs pedB $p = 0.0002$; *PROM1*-bulk ChP vs. adult $p = 0.0452$, ChP vs. pedB $p = 0.0122$; *BAIAP3*-bulk ChP vs. adult $p = 0.0177$, adult vs. pedB $p < 0.0001$; *WNT2B*-bulk ChP vs. adult $p = 0.0098$, adult vs. pedB $p < 0.0001$; *WNT5B*-bulk adult vs. pedB $p = 0.0016$; *GNB3*-bulk ChP vs. pedB $p = 0.0154$, adult vs. pedB $p = 0.0003$).

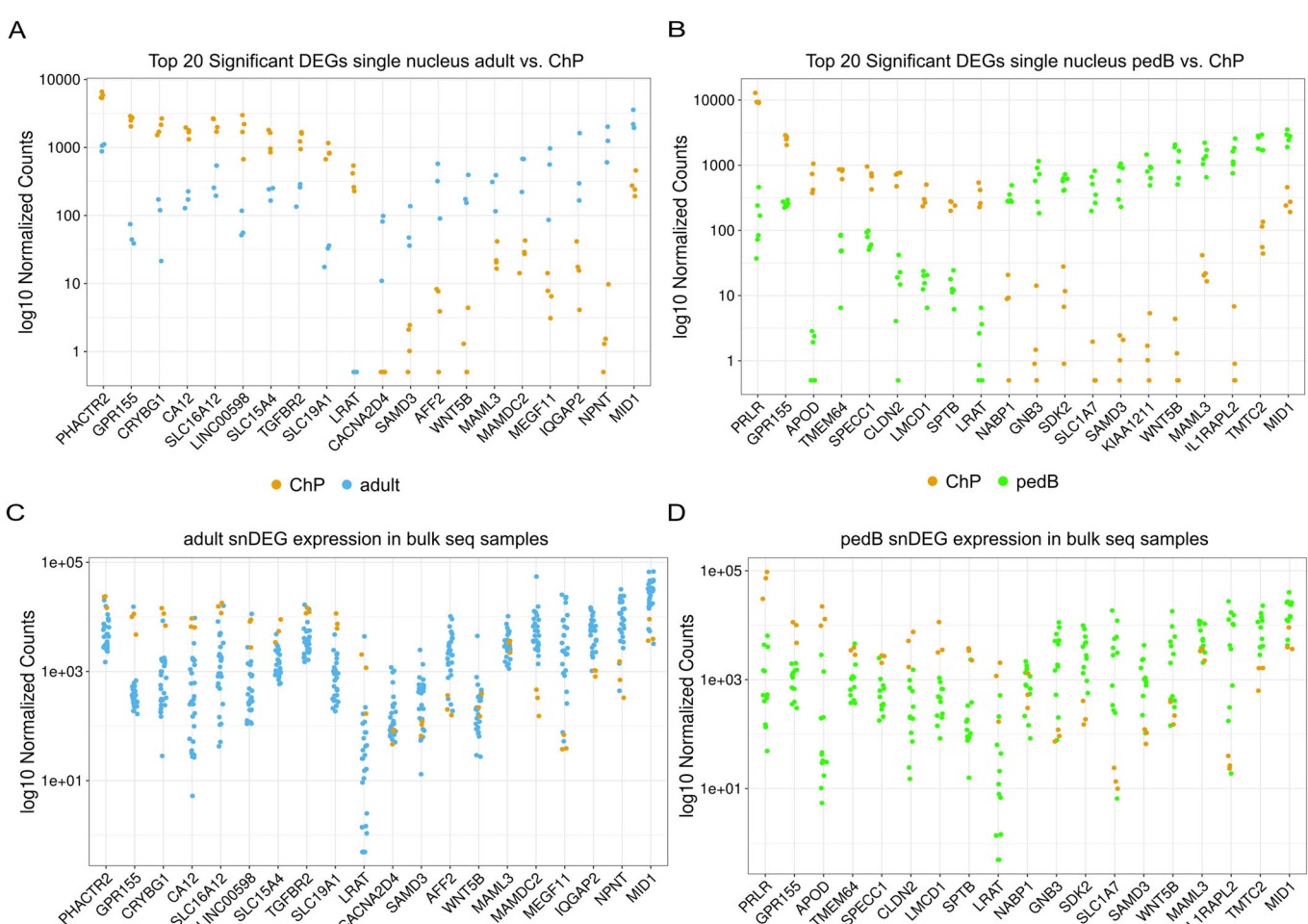

| Gene | L2FC (sn_adult) | L2FC (bs_adult) | Gene | L2FC (sn_pedB) | L2FC (bs_pedB) |
|---|---|---|---|---|---|
| PHACTR2 | -2.4* | -1.5* | PRLR | -5.7* | -4.8* |
| GPR155 | -5.5* | -2.8* | GPR155 | -3.3* | -2.4* |
| CRYBG1 | -4.1* | -3.0* | APOD | -8.8* | -4.5* |
| CA12 | -3.1* | -2.1 | TMEM64 | -3.7* | -1.2* |
| SLC16A12 | -2.6* | -1.9 | SPECC1 | -3.2* | -1.7* |
| LINC00598 | -4.5* | -2.1* | CLDN2 | -5.1* | -3.1* |
| SLC15A4 | -2.5* | -1.7* | LMCD1 | -4.1* | -3.4* |
| TGFBR2 | 2.5* | -1.2* | SPTB | -4.2* | -2.8* |
| SLC19A1 | -4.8* | -2.8* | LRAT | -7.8* | -1.3 |
| LRAT | -10.4* | -0.4 | NABP1 | 4.7* | 0.2 |
| CACNA2D4 | 10.11* | 0.7 | GNB3 | 9.4* | 5.2* |
| SAMD3 | 5.5* | 1.0 | SDK2 | 6.1* | 3.6* |
| AFF2 | 5.6* | 3.2* | SLC1A7 | 9.7* | 7.8* |
| WNT5B | 7.1* | 0.2 | SAMD3 | 8.8* | 3.3* |
| MAML3 | 3.5* | 0.1 | KIAA1211 | 8.5* | 4.3* |
| MAMDC2 | 3.9* | 4.3* | WNT5B | 9.6* | 3.5* |
| MEGF11 | 6.1* | 6.6* | MAML3 | 6.0* | 1.0* |
| IQGAP2 | 5.3* | 2.4* | IL1RAPL2 | 9.2* | 7.9* |
| NPNT | 8.3* | 3.4* | TMTC2 | 4.5* | 2.8* |
| MID1 | 3.1* | 2.1* | MID1 | 3.16* | 1.4* |

**Figure EV4.  Expression of 20 top differentially expressed genes in each methylation tumor profile.**

(**A–D**) Differentially expressed gene (DEG) expression in single nucleus (sn) and bulk sequencing (bs) samples. Normalized counts for 20 most significant DEGs for adult vs. disease-free choroid plexus (ChP) (**A**, **C**) or pedB vs. ChP (**B**, **D**) in each sn (**A**, **B**) or bs (**C**, **D**) library. (**E**) Expression fold change and adjusted *p*-values for genes graphed in (**A–D**) above. L2FC = log2 fold. * indicates significant DEGs (adjusted *p*-value < 0.05, log2 fold change >1 or <−1, Wald test-DESeq2).

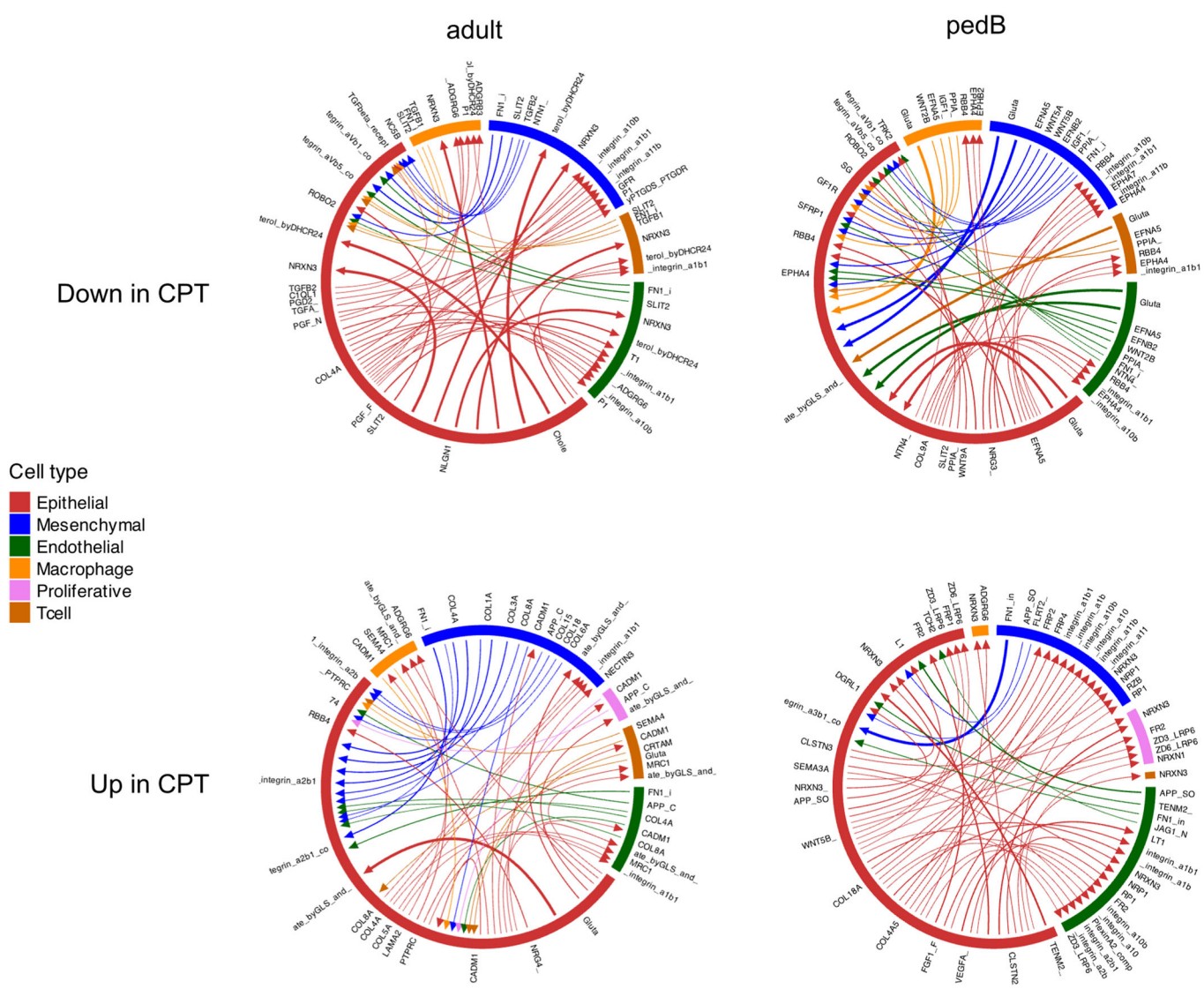

**Figure EV5.  Cell–cell interactions involving epithelial lineage cell differentially expressed genes in disease-free choroid plexus (ChP), adult, and pedB tumor samples.**

Circos interaction diagrams of top interactions in ChP (top row) or tumors (bottom row, adult on left, pedB on right). Interactions are limited to genes significantly downregulated in adult and pedB (top row) or upregulated in adult and pedB (bottom row).

