## [Peer Review File · The EMBO Journal]

Single-nucleus RNA-seq dissection of choroid plexus tumor cell heterogeneity

Anthony Hill, Konstantin Okonechnikov, Marla Herr, Christian Thomas, Supat Thongjuea, Martin Hasselblatt, and Annarita Patrizi

Corresponding authors: Anthony Hill (a.hill@dkfz-heidelberg.de) , Annarita Patrizi (a.patrizi@dkfz-heidelberg.de)

Review Timeline:

Submission Date:	7th Oct 23
Editorial Decision:	1st Dec 23
Revision Received:	17th Jun 24
Editorial Decision:	26th Sep 24
Revision Received:	8th Oct 24
Accepted:	15th Oct 24

Editor: Daniel Klimmeck

Transaction Report:

Dear Dr Annarita Patrizi, dear Dr Anthony Hill,

Thank you again for submitting your manuscript for consideration by the EMBO Journal. Please accept my sincere apologies for getting back to you with unusual protraction due to delayed referee input, as well as detailed discussion in the editorial team. Your manuscript has been seen by three referees, and we have received reports from all of them, which are shown below.

Given the referees' positive recommendations, I would like to invite you to submit a revised version of the manuscript, addressing the comments of all three reviewers. I should add that it is EMBO Journal policy to allow only a single round of revision, and acceptance of your manuscript will therefore depend on the completeness of your responses in this revised version.

I would appreciate if you could contact me during the next weeks for exchange e.g. a video call to discuss your perspective on the comments and potential plan for revisions.

Please feel free to contact me if you have any questions or need further input on the referee comments.

When submitting your revised manuscript, please carefully review the instructions below.

Please feel free to approach me any time should you have additional questions related to this.

Thank you for the opportunity to consider your work for publication.

I look forward to your revision.

Best regards,

Daniel Klimmeck

Daniel Klimmeck, PhD
Senior Editor
The EMBO Journal

Instruction for the preparation of your revised manuscript:

- 1) a .docx formatted version of the manuscript text (including legends for main figures, EV figures and tables). Please make sure that the changes are highlighted to be clearly visible.
- 2) individual production quality figure files as .eps, .tif, .jpg (one file per figure).
- 3) a .docx formatted letter INCLUDING the reviewers' reports and your detailed point-by-point response to their comments. As part of the EMBO Press transparent editorial process, the point-by-point response is part of the Review Process File (RPF), which will be published alongside your paper.
- 4) a complete author checklist, which you can download from our author guidelines ([https://wol-prod-cdn.literatumonline.com/pb-assets/embo-site/Author Checklist%20-%20EMBO%20J-1561436015657.xlsx](https://wol-prod-cdn.literatumonline.com/pb-assets/embo-site/Author%20Checklist%20-%20EMBO%20J-1561436015657.xlsx)). Please insert information in the checklist that is also reflected in the manuscript. The completed author checklist will also be part of the RPF.
- 5) Please note that all corresponding authors are required to supply an ORCID ID for their name upon submission of a revised manuscript.
- 6) It is mandatory to include a 'Data Availability' section after the Materials and Methods. Before submitting your revision, primary

datasets produced in this study need to be deposited in an appropriate public database, and the accession numbers and database listed under 'Data Availability'. Please remember to provide a reviewer password if the datasets are not yet public (see <https://www.embopress.org/page/journal/14602075/authorguide#datadeposition>).

7) Our journal encourages inclusion of *data citations in the reference list* to directly cite datasets that were re-used and obtained from public databases. Data citations in the article text are distinct from normal bibliographical citations and should directly link to the database records from which the data can be accessed. In the main text, data citations are formatted as follows: "Data ref: Smith et al, 2001" or "Data ref: NCBI Sequence Read Archive PRJNA342805, 2017". In the Reference list, data citations must be labelled with "[DATASET]". A data reference must provide the database name, accession number/identifiers and a resolvable link to the landing page from which the data can be accessed at the end of the reference. Further instructions are available at .

8) At EMBO Press we ask authors to provide source data for the main and EV figures. Our source data coordinator will contact you to discuss which figure panels we would need source data for and will also provide you with helpful tips on how to upload and organize the files.

Numerical data can be provided as individual .xls or .csv files (including a tab describing the data). For 'blots' or microscopy, uncropped images should be submitted (using a zip archive or a single pdf per main figure if multiple images need to be supplied for one panel). Additional information on source data and instruction on how to label the files are available at .

9) We replaced Supplementary Information with Expanded View (EV) Figures and Tables that are collapsible/expandable online (see examples in <https://www.embopress.org/doi/10.15252/emboj.201695874>). A maximum of 5 EV Figures can be typeset. EV Figures should be cited as 'Figure EV1, Figure EV2' etc. in the text and their respective legends should be included in the main text after the legends of regular figures.

11) For data quantification: please specify the name of the statistical test used to generate error bars and P values, the number (n) of independent experiments (specify technical or biological replicates) underlying each data point and the test used to calculate p-values in each figure legend. The figure legends should contain a basic description of n, P and the test applied. Graphs must include a description of the bars and the error bars (s.d., s.e.m.).

We realize that it is difficult to revise to a specific deadline. In the interest of protecting the conceptual advance provided by the work, we recommend a revision within 3 months (29th Feb 2024). Please discuss the revision progress ahead of this time with

the editor if you require more time to complete the revisions.

Referee #1:

Hill et al. report a single-cell transcriptomic atlas of healthy human choroid plexus, choroid plexus papilloma, and choroid plexus carcinoma. CPTs are rare tumors with few somatic driver gene mutations (TP53/TERT) and no therapeutically targetable alterations. The resource by Hill et al. is a valuable contribution to the molecular understanding of CPTs and will aid in the preclinical development of novel targeted therapies. The manuscript described cell types in the healthy/malignant choroid plexus tissues, chromosomal losses/gains in single cells, pathway enrichments in CPPs/CPCs, and computationally predicted cell-cell interactions.

Major comments

1. A major conclusion from the manuscript is that CPT methylation subtypes reflect differences in gene expression. Yet, no information is available in the Methods and Results section on DNA methylation subtyping of all 11 CPTs. The Introduction describes three CPT subtypes (Ped A, Ped B, Adult). Yet, the Heidelberg brain tumor classifier (eg, v12.5) defines four CPT subtypes (adult CPC, pediatric CPC, adult CPP, and pediatric CPP). The authors should present their results using the latest molecular brain tumor classification system. Table 1 should also report the calibrated DNA methylation classification score for all 11 CPTs.
2. CPTs have complex genomes and inference of chromosomal gains/losses from single-cell transcriptomes is challenging. It is unclear whether any statements are supported by the inference of chromosomal gains/losses from scRNA-seq data only (eg, that cells with many chromosomal losses/gains are more frequent in CPPs vs CPCs; robustness of evolutionary trees). DNA methylation arrays for the same cohort independently assess chromosomal gains/losses. The online version of the Heidelberg brain tumor classifier reports CNVs from DNA methylation arrays and the authors should use this data to assess the quality of single-cell-based CNV predictions (eg, frequent/clonal CNVs).
3. The authors used arbitrary cutoffs (>10 or >20 CNVs) to categorise genomic instability in CPTs. What was the rationale for these cutoffs? It would be more informative to show a boxplot/violin plot with the number of chromosomal gains/losses per tumor and assess the significance between CPP/CPC using statistical tests. Previous CPT studies with larger cohorts observed no differences in the total number of chromosomal gains/losses between molecular CPT subgroups (Thomas et al. Neuro Oncol 2021).
4. Relative quantification of cell types with droplet-based single-cell RNA sequencing can result in strong biases due to differential capture rates of specific cell populations. A more robust approach is to derive cell-type marker genes from single-cell the CPT atlas and perform cell-type deconvolution using bulk RNA sequencing of CPTs. This would validate that macrophages and endothelial cells are enriched in CPTs (eg, using RNA-seq data derived from the same CPT cohort; RNA-seq data from the Children's Brain Tumor Network; or RNA-seq data from Thomas et al. Neuro-Oncology 2021. PMID: 33249490). Notably, no quantification of Iba1-positive cells was described in tissue sections of CPTs and healthy controls.

Minor comments

1. A table with information about quality control metrics for all samples is missing (# of cells per sample, mean # of reads per cell, # genes per cell, total number of reads per sample).
2. MAPK and PI3K pathway enrichment in CPTs are potentially therapeutically relevant. It would be informative if the authors could describe in more detail the set of genes which contributed to these enrichments. Are these pathways enriched in all studied CPTs or only a subset of CPTs? To what extent are these and similar pathway enrichments driven by single samples that are represented by many cells in this atlas?
3. Figure 3B suggests that some healthy ChPs can also present with many chromosomal gains/losses. Were these normal cells classified as epithelial?
4. The authors state in the Methods section that the scRNA-seq dataset will be uploaded to GEO. The manuscript currently provides no GEO accession number. The data upload should include raw scRNA-seq counts, processed sc-RNA-seq data, and raw DNA methylation arrays (IDAT files).
5. The Methods section on Iba1 IHC contains no information about the supplier and clone.

Referee #2:

Authors Hill et al. submit a manuscript titled "Single Nucleus RNA-seq dissection of choroid plexus tumor cell heterogeneity" for consideration in EMBO. The authors use single nucleus RNA sequencing analysis of human choroid plexus tumors and normal brains. They report cellular profiles, transcriptional heterogeneity, putative cell interaction networks, and copy number alterations. These analyses suggest changes in epithelial gene transcription and genome wide methylation profile changes along with macrophage and mesenchymal gene expression changes. Overall, these studies present a valuable resource for investigators and authors are commended on including normal and two different cancer states. However, no validation of marker predictions

or cell signaling interactions are presented despite several tools now being standardized for human tissue analysis. Several opportunities are available to validate interesting observations reported in Figure 2D, Figure 4C, D, and Figure 7. These data will enrich the resource and provide valuable guidelines for future histopathological analyses of choroid plexus. Finally, as part of minor comments Figure 3B needs revisions or summary diagrams to guide the reader.

Referee #3:

In the manuscript "Single Nucleus RNAseq dissection of choroid plexus tumor cell heterogeneity" the authors perform snRNA-seq of human primary choroid plexus and choroid plexus tumors. Authors investigate single nucleus gene expression from human choroid plexus (N=4) and 11 choroid plexus tumor samples (N=4 CPP, N=2 aCPP, N=5 CPC) to investigate common pathways underlying choroid plexus tumors. These samples of rare tumors are valuable and data gathered will serve as a useful resource for understanding tumor initiation and tumor heterogeneity in the choroid plexus.

The experiments are logical, and the samples and single cell perspective are valuable. However, in its current state, it is difficult to interpret the findings of the manuscript due to missing quality control (QC) data and data analysis, as well as poor data visualization methods. Many of the claims in the text are not well supported by the data shown, as statistical measures are rarely shown, or the authors do not perform the proper analyses to test the statements they have made. Overall, while the underlying dataset may be useful, more work should be done to appropriately analyze and present the data in the best possible way.

In terms of novelty, this study contributes important updates to human choroid plexus cell atlases and gene expression in choroid plexus tumors. Authors should, however, more fully address relevant citations in the background and contextualize their findings in the field.

This manuscript generates a new snRNA-seq dataset for an important rare pediatric tumor type, investigates how subtypes of choroid plexus tumors (CPP, aCPP, CPC) differ from one another largely along the choroid plexus epithelial cell axis, and where they are similar. It also probes CNVs that could help additionally stratify patient populations. The study is relevant to the readership of EMBO and, once the QC is assessed, background updated, and analysis of vasculature is included, it has the potential to be useful to the field.

Recommendation: Major Revision

Specific comments are below.

Major Concerns:

1. Data quality control: The claims of the manuscript cannot be evaluated without additional QC information for snRNA-Seq. This information should be added to the supplemental figures. Major additions should be made to the methods section to detail data processing methods, including code used to process the data. QC information such as reads / cell, genes / cell, % mitochondrial reads, cells / cluster, cells / sample / cluster, etc. should all be provided in the supplement to provide confidence to the reader that the underlying data quality is good and not driven solely by batch effects from individual samples. More presentations by sample would help assure the readers. In its current state, this information cannot be gleaned from the tSNE plots provided, and the tSNE-plots are not as informative as this information would be if more simply presented in a bar graph (such as percent of cells / sample / cluster).

a. Any modern guide for single cell QC would be appropriate to follow, however, the authors might refer to one published in EMBO here.

i. <https://www.embopress.org/doi/full/10.15252/msb.20188746>

2. Additional relevant literature should be introduced in the background including:

a. Tong, et al. *Cancer Cell*. 2015 [PMID: 25965574]

b. Shannon, et al. *Am J Pathol*. 2018 [PMID: 29545198]

c. Li, et al. *Cell Death Differentiation*. 2022 [PMID: 35322202]

3. Methods: In general, methods are too sparse to be useful. A few examples below, but they are not exhaustive.

a. It is unclear whether all the tissues were prepared the same way - frozen, FFPE, fresh, storage conditions etc.

b. Antibody information is missing.

c. Data analysis methods should be more complete. Scripts used to analyze the data should be deposited in a publicly accessible repository to enable reproducible data analysis.

4. Data availability:

a. I am unable to review data availability because no reviewer key for the GEO data is available.

5. Validation:

a. Key identities for claims of epithelial, immune or mesenchymal diversity should be validated in tissue.

b. A figure element comparing the current tumor gene expression with published canonical CPP/CPC gene expression (TP53, MYC, Notch, PTEN, TAF12, NFYC, and RAD54L) would help contextualize the data within the field.

6. Many of the claims in the paper could be much better understood if the authors provided more comprehensive data analysis. For example, the authors state with regards to Figure 1: "In brief, in the healthy ChP samples, CPECs from younger and older individuals cluster separately, perhaps reflecting age dependent transcriptional changes observed in murine CPECs[14]." The authors have the data to determine the differential genes between younger and older CPECs in healthy ChP samples, therefore they should provide the analysis and check if those genes match with the ones identified in previous studies, or if they are novel genes.

7. While inferring CNV is interesting, the data visualization and methods in Figure 3 require revision. The authors themselves state that it was difficult to find any stereotyped pattern from the phylogenetic dendrogram, and the inclusion of 3D does not seem that useful. In addition, much of the analysis in 3A-3C would benefit from statistical confidence. The authors make confusing claims in the text, such as "Common CNVs in CPP involved gain of 4q, 7p, 7q, and 12q", however, based on data provided in 3A, other CNVs appear as involved as the ones called out in the text. Overall, the data presentation in this figure should be improved to make it easier to see the conclusions, and statistical measures should be provided.

a. Authors should also clarify how their CNVs relate to those previously discovered in their Ref 6 Thomas et al 2016 (i.e., which cluster is closest to PedB?)

8. For many of the dotplots presented in this paper, it is unclear how the marker genes were selected, and if these are the top marker genes determined computationally, or if these are a curated list. In addition, there is often a lack of marker genes in the dotplots present to understand why two clusters are different (for example, in 1D, it is unclear what separates CC2-Epi & CC3-Epi or in 2D it is unclear what separates epithelial1 & epithelial2).

9. Vasculature: The authors mention that endothelial cells were present including arterial and venous components. However, expanded analysis for these cell types would be informative to considering vasculature as a therapeutic target and should be readily available from the data collected here.

Minor Concerns:

1. Introduction:

a. Cite references the following claims "Overexpression of Myc in the murine CNS, alone or in combination with Tp53 deletion, causes CPT in transgenic mice".

2. Statistics:

a. The number of individual samples should be stated in the Abstract, as that information is more useful than the number of cells.

b. Please clarify why FDR of 0.1 rather than 0.05 was chosen.

c. Discuss whether any differences associated with LV choroid plexus vs 4V choroid plexus are discoverable in the diverse samples and if so, what proportion of the differences are driven by areal components?

3. Discussion: The cell type diversity is a key strength of this manuscript, and a more detailed discussion of vasculature, endothelial cells, and maturity is warranted.

4. Figures: Figure quality is poor throughout including:

a. A number of cropping artifacts.

b. Distortions from resizing.

c. Color schemes change throughout - i.e. within Figure 7B.

d. Scales for dot plots are not always provided.

We would like to thank the reviewers for their thoughtful and critical consideration of the manuscript. Following the guidance of their constructive comments, we have prepared what we believe to be a greatly improved revised manuscript. We hope that with the additions and changes made our paper is now acceptable for publication in *The EMBO Journal*. Changes in the revised text documents have been highlighted with colored text (light blue regular).

Please find our point-by-point responses to the reviewers' comments below. We will cite the reviewers' comments in full in italic typeface and provide our responses in light blue regular typeface.

Reviewer #1

Major comments:

1. *A major conclusion from the manuscript is that CPT methylation subtypes reflect differences in gene expression. Yet, no information is available in the Methods and Results section on DNA methylation subtyping of all 11 CPTs. The Introduction describes three CPT subtypes (Ped A, Ped B, Adult). Yet, the Heidelberg brain tumor classifier (eg, v12.5) defines four CPT subtypes (adult CPC, pediatric CPC, adult CPP, and pediatric CPP). The authors should present their results using the latest molecular brain tumor classification system. Table 1 should also report the calibrated DNA methylation classification score for all 11 CPTs.*

We thank the reviewer for raising this point. In the revised Table 1, we have now included detailed information regarding DNA methylation profiling of all sequenced CPTs. Alongside the classification results from the initial v11b4 release (based on samples from our previous study, Thomas et al. *Neuro. Oncol.*, PMID: 26826203), we have also incorporated data from the most recent local v12.8 Classifier version. The key difference between these versions lies in the division of the previous "pediatric B" group

into two age-related subgroups: 1) “choroid plexus carcinoma, pediatric subtype” and 2) “choroid plexus carcinoma, adult subtype”. Notably, our adult CPC sample (#CC3) and an adolescent aCPP sample (#aCP2) were categorized into the latter group.

However, we believe it is important to mention that these two groups are currently under closer examination and are considered subgroups of the original “pediatric B” cluster. Additionally, we find that the term “choroid plexus carcinoma” for a methylation-defined group could be misleading and may cause confusion, as some aCPP and CPP samples can also be classified into this group. We, therefore, prefer the more neutral term “pediatric B”.

Given the relatively small sample size, we have decided to adhere to the v11b4 subgroup assignment, which essentially encompasses the two “choroid plexus carcinoma” subgroups.

- 2. CPTs have complex genomes and inference of chromosomal gains/losses from single-cell transcriptomes is challenging. It is unclear whether any statements are supported by the inference of chromosomal gains/losses from scRNA-seq data only (eg, that cells with many chromosomal losses/gains are more frequent in CPPs vs CPCs; robustness of evolutionary trees). DNA methylation arrays for the same cohort independently assess chromosomal gains/losses. The online version of the Heidelberg brain tumor classifier reports CNVs from DNA methylation arrays and the authors should use this data to assess the quality of single-cell-based CNV predictions (eg, frequent/clonal CNVs).*

We thank the reviewer for this comment. In order to verify the observed CNV analysis results from single nucleus data, we first inspected the CNVs derived from bulk DNA methylation profiles for each case from our target cohort (Rebuttal Figure 1, below). Afterwards, we used this information to assess the detection quality of CNVs derived from single nucleus RNA-seq per sample (see revised Figure 3A). Overall, identified CNVs from single nucleus data were matched compared to CNVs from DNA methylation for each tumor sample. For example, from the visual inspection in most of the samples that have a high number of cells (aCP1, CC2, CC3, CC4, CC5, and CP2), CNV results are matched well between the two approaches. Also, these comparison results demonstrated that stable amplification/deletion patterns within a sample are present across all tumor cells (see revised Figure 3B).

Rebuttal Figure 1. Regions of genomic gain (green) and loss (red) in CPT samples as assessed by methylation profiling chip signal.

3. *The authors used arbitrary cutoffs (>10 or >20 CNVs) to categorise genomic instability in CPTs. What was the rationale for these cutoffs? It would be more informative to show a boxplot/violin plot with the number of chromosomal gains/losses per tumor and assess the significance between CPP/CPC using statistical tests. Previous CPT studies with larger cohorts observed no differences in the total number of chromosomal gains/losses between molecular CPT subgroups (Thomas et al. Neuro Oncol 2021).*

The rationale for cutoffs was based on inspection of CNV distribution signal. When we compared the mean number of CNV gains derived from single nucleus data between adult and pedB we observed a small but statistically significant variance between them when inspecting only gains (Rebuttal Figure 2A, below). However, no significant variance was identified when inspecting only losses (Rebuttal Figure 2B, below) and full CNV count (gains + losses) (Rebuttal Figure 2C, below). We, therefore, decided to mainly report most common CNVs confirmed by the methylation profiling data (see revised Figure 3).

Rebuttal Figure 2. Mean CNV counts per tumor group. Counts of gains (A), losses (B) or both (C) in adult (blue) or pedB (gold) samples.

4. *Relative quantification of cell types with droplet-based single-cell RNA sequencing can result in strong biases due to differential capture rates of specific cell populations. A more robust approach is to derive cell-type marker genes from single-cell the CPT atlas and perform cell-type deconvolution using bulk RNA sequencing of CPTs. This would validate that macrophages and endothelial cells are enriched in CPTs (eg, using RNA-seq data derived from the same CPT cohort; RNA-seq data from the Children's Brain Tumor Network; or RNA-seq data from Thomas et al. Neuro-Oncology 2021. PMID: 33249490). We agree with the reviewer that this is a critical point of our dataset. We, therefore, performed deconvolution using adult and pedB bulk sequencing files from Thomas et al., as suggested (see revised Figure 5E), confirming decreased mesenchymal cell numbers in adult and pedB samples, as well as increased proliferative and endothelial cells in pedB samples. The changes in pedB cell numbers were each significant at the adjusted p-value < 0.01 level. We also used RT-qPCR to test for expression of four genes previously identified as mesenchymal cell markers (ADAM12, VCAM, ANGPT1 and LAMC3 Dani et al. Cell 2021, PMID: 33932339; Yang et al Nature 2021, PMID: 34153974; Toma et al. eNeuro 2020, PMID 32349983) in a small cohort (3 CPP plus 3 CPC) of CPT samples. The size of the used cohort is correlated to the limited availability of freshly frozen CPT samples. We observe a trend towards decreased expression in CPP and CPC samples for*

all selected markers, but only *ANGPT1* expression for CPC vs. ChP reached significance (see revised Figure 5F).

Notably, no quantification of Iba1-positive cells was described in tissue sections of CPTs and healthy controls.

Due to the lack of available choroid plexus disease-free control samples preserved in formalin, we only performed a detailed analysis of Iba1-positive cells in CPP and CPC samples. Our data showed no differences in the total Iba1 cell density between the two tumors but a significant increase of the number of Iba1-positive ramifications in CPC compared to CPP sections (see revised Figure 6D-G).

Minor comments:

A. A table with information about quality control metrics for all samples is missing (# of cells per sample, mean # of reads per cell, # genes per cell, total number of reads per sample).

We thank the reviewer for this important comment. We, now, added a table including # of cells, # of RNA, # of genes and percentage of mitochondria per each sample, samples grouped by methylation profile and samples grouped by on histological tumor classification (see Appendix Table S1).

B. MAPK and PI3K pathway enrichment in CPTs are potentially therapeutically relevant. It would be informative if the authors could describe in more detail the set of genes which contributed to these enrichments. Are these pathways enriched in all studied CPTs or only a subset of CPTs? To what extent are these and similar pathway enrichments driven by single samples that are represented by many cells in this atlas?

An extended analysis of all identified MAPK- and PI3K-related genes showed enrichment of one or both of these pathways within most of the analyzed tumor samples. Revised Figure 4G shows per sample expression for 42 of the most highly expressed DEGs in the MAPK and/or PI3K signaling pathways. While the magnitude and number of upregulated genes varies, there is a clear upregulation of both pathways in most pedB samples. We also detected upregulation of a smaller subset of PI3K genes in adult profile samples.

C. Figure 3B suggests that some healthy ChPs can also present with many chromosomal gains/losses. Were these normal cells classified as epithelial?

All of the cells in the previous submitted version of Figure 3B were epithelial. We detected a small number (<10) of CNS in a few

percent of disease free ChP cells. This reference visualization is now included in revised Figure 3A.

D. The authors state in the Methods section that the scRNA-seq dataset will be uploaded to GEO. The manuscript currently provides no GEO accession number. The data upload should include raw scRNA-seq counts, processed sc-RNA-seq data, and raw DNA methylation arrays (IDAT files).

Raw and processed human choroid plexus and human choroid plexus tumor sequencing files are available at the Gene Expression Omnibus (GEO) database GSE264154. See Data and Code availability section on page 17.

E. The Methods section on Iba1 IHC contains no information about the supplier and clone.

Catalog number, RRID identified code, dilution and supplier information for anti-Iba1 have been added to materials and methods; page 16.

Referee #2:

Major comment:

Overall, these studies present a valuable resource for investigators and authors are commended on including normal and two different cancer states. However, no validation of marker predictions or cell signaling interactions are presented despite several tools now being standardized for human tissue analysis. Several opportunities are available to validate interesting observations reported in Figure 2D, Figure 4C, D, and Figure 7. These data will enrich the resource and provide valuable guidelines for future histopathological analyses of choroid plexus.

We thank the reviewer for this critical comment. We've now used a combination of previously published bulk sequencing data and RT-qPCR to confirm DEGs identified in our single nucleus data set. New Figure EV3 shows expression changes for 6 epithelial cell DEGs identified in the snRNAseq pseudobulk analysis (Figure EV3A). RT-qPCR (Figure EV3B) and bulk sequencing (Figure EV3C) confirmed the average expression changes of these genes in adult and/or pedB tumors. While the small number of samples used for RT-qPCR do not reach significance, the larger bulk sequencing data set shows significant differences (Figure EV3C) for most of the validated genes. In New Figure EV4 we look at expression of the 20 most significant DEGs in adult (Figure EV4A,C,E) and pedB (Figure EV4B,D,E) tumors. The direction of expression changes in our snRNAseq dataset (i.e. positive or negative log₂FoldChange) predicts the direction of change observed in the bulk sequencing data (Figure EV4E) for 100% of the

genes tested. These expression changes exceed our significance criteria ($\log_2\text{FoldChange} \geq 2$, adjusted p-value < 0.05) for 13/20 adult profile DEGs and 18/20 pedB DEGs. We also used RT-qPCR and previously published bulk sequencing data to confirm the reduction of mesenchymal markers in adult and pedB profile data (revised Figure 5E-F) and macrophage markers in pedB profiles (revised Figure 6B-C).

Minor comment:

Finally, as part of minor comments Figure 3B needs revisions or summary diagrams to guide the reader.

We thank the reviewer for this constructive criticism. In the revised Figure 3 we focus on sample counts of the most common CNVs in adult and pedB samples, and present them in a way that we feel is clear and intuitive.

Referee #3:

Major comments:

1. *Data quality control: The claims of the manuscript cannot be evaluated without additional QC information for snRNA-Seq. This information should be added to the supplemental figures. Major additions should be made to the methods section to detail data processing methods, including code used to process the data. QC information such as reads / cell, genes / cell, % mitochondrial reads, cells / cluster, cells / sample / cluster, etc. should all be provided in the supplement to provide confidence to the reader that the underlying data quality is good and not driven solely by batch effects from individual samples. More presentations by sample would help assure the readers. In its current state, this information cannot be gleaned from the tSNE plots provided, and the tSNE-plots are not as informative as this information would be if more simply presented in a bar graph (such as percent of cells / sample / cluster).*

To improve data visualization and interpretation, we have now added per sample cellular composition (Expanded Figure S2B) and gene expression (revised Figure 4, Expanded Figure 4, Appendix Figure 1).

- a. *Any modern guide for single cell QC would be appropriate to follow, however, the authors might refer to one published in EMBO here. <https://www.embopress.org/doi/full/10.15252/msb.20188746>*
We have now assembled cell number, average UMI/cell, average # genes detected, and percent mitochondrial reads per sample in Appendix Table S1. See also answer to reviewer #1, minor comment a.

2. *Additional relevant literature should be introduced in the background including:*
 - a. Tong, et al. *Cancer Cell*. 2015 [PMID: 25965574]
 - b. Shannon, et al. *Am J Path*. 2018 [PMID: 29545198]
 - c. Li, et al. *Cell Death Differentiation*. 2022 [PMID: 35322202]

These references have been added to the introduction; page 3.

3. *Methods: In general, methods are too sparse to be useful. A few examples below, but they are not exhaustive.*
 - a. *It is unclear whether all the tissues were prepared the same way - frozen, FFPE, fresh, storage conditions etc.*
 snRNAseq and qRT-PCR samples were frozen and stored at -80C. Sections used for immunohistochemistry staining were, instead, formalin fixed. All these information are now clearly stated within the appropriate sections of the materials and methods. In addition, we now added Appendix Table S9 including sample information for tissue used for immunohistochemistry.
 - b. *Antibody information is missing.*
 See answer to reviewer #1, minor comment e.
 - c. *Data analysis methods should be more complete. Scripts used to analyze the data should be deposited in a publicly accessible repository to enable reproducible data analysis.*
 Data analysis methods have now been extended in the materials and methods for all described techniques; pages 16-17. Scripts have been made available at https://github.com/corticaltone/CPT_Atlas.

4. *Data availability:*
 - a. *I am unable to review data availability because no reviewer key for the GEO data is available.*
 See answer to reviewer #1, minor comment D.
 Shortly, raw and processed data files have been uploaded to the Gene Expression Omnibus database (GSE264154).

5. *Validation:*
 - a. *Key identities for claims of epithelial, immune or mesenchymal diversity should be validated in tissue.*
 Thank you for such a critical point. We do agree that validation in tissue is a critical step. Due to the limited amount of available samples and difficulties of identified reliable and specific antibodies for many selective markers, we decided to implement the use of available freshly frozen CPT samples and used RT-qPCR to perform targeted mRNA gene expression analysis of selective markers for epithelial (see New Figure EV3B), immune (see revised Figure 6B) and mesenchymal markers (see revised Figure 5F). In addition, we also validated significant marker genes and DEGs

using normalized gene counts from published bulk sequencing data (see New Figure EV3C for epithelial markers; see revised Figure 6C for immune markers and see revised Figure 5E for mesenchymal markers). Finally, detailed histochemical analyses of Iba1 positive macrophages have also been included (see revised Figure 6D-G).

b. A figure element comparing the current tumor gene expression with published canonical CPP/CPC gene expression (TP53, MYC, Notch, PTEN, TAF12, NFYC, and RAD54L) would help contextualize the data within the field.

Gene expression changes and adjusted p-values for these genes have been included in Appendix Table S2. To note, Notch and Pten regulated pathways (Notch signaling and PI3K-AKT signaling) are among the significant KEGG pathways activated in both adult and pedB tumor samples (Appendix Table S4). In addition, we now included an extended analysis of all identified PI3K-AKT-related genes organized by sample in revised Figure 4G.

6. *Many of the claims in the paper could be much better understood if the authors provided more comprehensive data analysis. For example, the authors state with regards to Figure 1: "In brief, in the healthy ChP samples, CPECs from younger and older individuals cluster separately, perhaps reflecting age dependent transcriptional changes observed in murine CPECs[14]." The authors have the data to determine the differential genes between younger and older CPECs in healthy ChP samples, therefore they should provide the analysis and check if those genes match with the ones identified in previous studies, or if they are novel genes.*

Thank you to the reviewer for this significant comment. Due to the limited tissue availability our healthy ChP samples contain 2 young adult, 1 middle aged, and 1 geriatric sample (see revised Table 1). We do not believe that the current set of disease-free ChP samples allow us to address normal transcriptional changes in the aging choroid plexus. A recent published paper demonstrated significant differences in the transcriptomic profiles across ages and cell types in murine ChPs (Dani et al., Cell 2021, PMID: 33932339). Due to the importance of this topic and the limited availability of healthy ChP samples for expanding our experimental cohort, we decided to not focus on this initial result and to not include this preliminary information in the current manuscript.

7. *While inferring CNV is interesting, the data visualization and methods in Figure 3 require revision. The authors themselves state that it was difficult to find any stereotyped pattern from the phylogenetic dendrogram, and the inclusion of 3D does not seem that useful. In addition, much of the analysis in 3A-3C would benefit*

from statistical confidence. The authors make confusing claims in the text, such as "Common CNVs in CPP involved gain of 4q, 7p, 7q, and 12q", however, based on data provided in 3A, other CNVs appear as involved as the ones called out in the text. Overall, the data presentation in this figure should be improved to make it easier to see the conclusions, and statistical measures should be provided.

We are grateful to the reviewer for these critical comments. In order to avoid confusion and highlight the main results, we redesigned Figure 3 (see revised Figure 3). Now, it starts from the main single nucleus results: from the InferCNV analysis of tumors with normal cells included as reference control. This figure demonstrates that there is stability in gain/loss patterns across all tumor cells per sample (revised Figure 3A). We also verified all the observations per sample by comparison to bulk DNA methylation CNV profiles (Rebuttal Figure 1). Overall, revised Figure 3 focuses on general inspection gains/losses between CPP and CPC.

A precise statistical comparison showed no significant differences when comparing the total amount of CNV changes derived from single nucleus data between CPP and CPC (see answer to reviewer # 1, major comments 4, and Rebuttal Figure 2A-C). Overall, this result did not demonstrate any group specific enrichment of CNVs.

Also, we're sorry for the confusion in the statement about common CNV. In order to state a clear message, we collected the statistics across CNVs and provided possible visualization showing this distribution in our single nucleus data cohort (see revised Figure 3B). This result demonstrated general enrichment of gains in full chromosomes 12, 1 as well as 20p most enriched in our cohort.

We also agree with the reviewer that initial Figure 3D showed strong sample specificity, therefore the phylogenetic dendrogram was removed from the manuscript.

a. Authors should also clarify how their CNVs relate to those previously discovered in their Ref 6 Thomas et al 2016 (i.e., which cluster is closest to PedB?)

Our sample cohort is a subset of samples previously analyzed in Thomas et al., 2021 (PMID: 33249490). The methylation classification used in Thomas et al., 2021 is based on previously published data (Thomas et al., 2016, PMID: 26826203). We performed direct comparison of our data to Thomas et al 2021 by inspecting CNV properties for adult and pedB separately (see Rebuttal Figure 1, above). Compared to Thomas et al., 2021, the most common pedB profile CNVs inferred from our data set include gain of 1p, 1q, 4p, 4q, 12p, 12q and 20p, each of which was frequently observed for pedB profile samples in Thomas et al 2021. For adult profile tumors, we observe frequent gain of chromosome arms 9p and 9q, along with loss of 21p and 21q, also consistent with

Thomas et., 2021, except for 21q loss, which was not reported previously (revised Figure 3). Differences between these two datasets are most likely due to the different size of the analyzed cohort (11 vs 47). Revised Figure 3 is now focused on presenting the most common CNVs per methylation profile type.

8. *For many of the dotplots presented in this paper, it is unclear how the marker genes were selected, and if these are the top marker genes determined computationally, or if these are a curated list. In addition, there is often a lack of marker genes in the dotplots present to understand why two clusters are different (for example, in 1D, it is unclear what separates CC2-Epi & CC3-Epi or in 2D it is unclear what separates epithelial1 & epithelial2).*

All the dotplots have been updated to present the lowest p-value marker genes for each cluster. In a few instances well characterized cell type specific markers (*OTX2*, *PECAM1*, *VCAM1*, etc.) were used, along with citation from the literature.

9. *Vasculature: The authors mention that endothelial cells were present including arterial and venous components. However, expanded analysis for these cells types would be informative to considering vasculature as a therapeutic target and should be readily available from the data collected here.*

In batch corrected data from disease-free choroid plexus, we detect a cluster expressing arterial cell markers and a second cluster expressing venous and capillary associated markers. A brief summary of these cell types has been added to the discussion on page 13. To note, in integrated data sets containing tumor samples endothelial marker positive cells form only a single cluster in batch corrected (Revised Figure 2B-C, Extended Figure 2A) and uncorrected (Revised Figure 1C-D) data. Perhaps due to the limited number of samples in this study, relatively few significant DEGs were detected in CPT endothelial cells (Appendix Table S8). This prevented a detailed analysis of pathways altered in endothelial cells.

Minor Concerns:

A. *Introduction:*

a. *Cite references the following claims "Overexpression of Myc in the murine CNS, alone or in combination with Tp53 deletion, causes CPT in transgenic mice".*

References were added to the text, including Merve et al. 2019 (PMID: 31142360); Shannon et al. 2018 (PMID: 29545198); Wang et al. 2019 (PMID: 30885981); page 3.

B. *Statistics:*

a. *The number of individual samples should be stated in the Abstract, as that information is more useful than the number of cells. Opening sentence in the Abstract has been changed as follows “To examine the cellular and transcriptional heterogeneity of choroid plexus tumors we determined the single nucleus transcriptomes of 23,906 nuclei from 4 disease-free choroid plexus and 11 choroid plexus tumors.”*

b. *Please clarify why FDR of 0.1 rather than 0.05 was chosen.*

We agree that the more conservative value 0.05 should be used. The manuscript and figures have been revised to use an FDR of 0.05 throughout.

c. *Discuss whether any differences associated with LV choroid plexus vs 4V choroid plexus are discoverable in the diverse samples and if so, what proportion of the differences are driven by areal components?*

The ventricle location of the healthy ChP samples was recorded for only 2 of the 4 sequenced samples. Similarly, the tumor sample location was only available for a subset of the sequenced samples (7 out of 11 samples). Of those, only 1 is listed as occurring in the 4th ventricle, 7 are listed as LV, and 1 is listed as parietal left (see revised Table 1). Therefore, it was not possible for us to address this specific question.

C. *Discussion: The cell type diversity is a key strength of this manuscript, and a more detailed discussion of vasculature, endothelial cells, and maturity is warranted.*

We agree with the reviewer that expanding our understanding of cellular diversity within endothelial cells and aging would be critical. We have added a brief summary of endothelial cell types in disease-free ChP to the discussion on page 13. However, because the manuscript is primarily focused on changes in CPT and not on characterizing normal ChP this discussion is not extensive. In addition, as mentioned in answers 6 and 9, study design and data collected do not give a strong basis for discussing either ChP maturation or endothelial cell states in CPT.

D. *Figures: Figure quality is poor throughout including:*

a. *A number of cropping artifacts.*

b. *Distortions from resizing.*

c. *Color schemes change throughout - i.e. within Figure 7B.*

d. *Scales for dot plots are not always provided.*

All figures have been redone with careful attention to the above defects.

On behalf of all authors,

Annarita Patrizi, PhD Independent Group Leader, German Cancer Research Center (DKFZ), Heidelberg, Germany

Anthony D. Hill, PhD, postdoctoral fellow, German Cancer Research Center (DKFZ), Heidelberg, Germany.

Dear Dr Patrizi, dear Dr Hill,

Thank you for submitting your revised manuscript (EMBOJ-2023-115823R) to The EMBO Journal. Your amended study was sent back to the three referees for their scientific re-evaluation, and we have received detailed comments from one of them, which I enclose below. As you will see, the expert states that the work has been substantially improved by the revisions and s/he is now broadly in favour of publication. Please note that we have editorially assessed your response to the other referees and found the concerns to be addressed satisfactorily.

Thus, we are pleased to inform you that your manuscript has been accepted in principle for publication in The EMBO Journal.

We now need you to take care of a number of issues related to formatting and data presentation as detailed below, which should be addressed at re-submission.

Please contact me at any time if you have additional questions related to below points.

As you might have seen on our web page, every paper at the EMBO Journal now includes a 'Synopsis', displayed on the html and freely accessible to all readers. The synopsis includes a 'model' figure as well as 2-5 one-short-sentence bullet points that summarize the article. I would appreciate if you could provide this figure and the bullet points.

Thank you for giving us the chance to consider your manuscript for The EMBO Journal. I look forward to your final revision.

Again, please contact me at any time if you need any help or have further questions.

Best regards,

Daniel Klimmeck

>> Please limit the number of keywords for your study to maximally five.

>> Author Contributions: Please remove the author contributions information from the manuscript text. Note that CRediT has replaced the traditional author contributions section as of now because it offers a systematic machine-readable author contributions format that allows for more effective research assessment. and use the free text boxes beneath each contributing author's name to add specific details on the author's contribution.

More information is available in our guide to authors.

>> Adjust the title of the 'Disclosure Statement' section to 'Disclosure and Competing Interests Statement'.

>> Funding: is a project number available for the funding? If so, please add to our manuscript online system and the Ackn section.

>> Dataset EV legends: The file "appendix table s3" should be renamed "Dataset EV1" .

>> Supplemental figures should be renamed "Figure EV1" in the legends.

>> Data availability section: Please remove the referee tokens and make sure data privacy is released. Add a URL for the GEO dataset.

>> Consider additional changes and comments from our production team as indicated below:

- DAS:

Please note that the specific URL for GSE264154 dataset is not provided in the data availability statement.

- Figure legends:

1. Please note that the exact p values are not provided in the legends of figures 4c-d; 5f; 6b; EV 3c.
2. Please indicate the statistical test used for data analysis in the legends of figures 4c-d; EV 2b; EV 3a; EV 4e.
3. Please note that in figure EV 3d; there is a mismatch between the annotated p values in the figure legend and the annotated p values in the figure file that should be corrected.
4. Please note that the box plots need to be defined in terms of minima, maxima, centre, bounds of box and whiskers, and percentile in the legends of figures 5e; 6c; EV 2d.
5. Please note that information related to n is missing in the legends of figures 5e; 6c; EV 1a-b; EV 2d.

Referee #3:

We thank the authors for a thorough and responsive revision process. All suggestions have been addressed in the updated text, figures, and appendices. I look forward to seeing this work published for the readership of EMBO.

The authors addressed the minor editorial issues.

Dear Dr Annarita Patrizi, dear Dr Anthony Hill,

Thank you for submitting the revised version of your manuscript. I have now evaluated your amended manuscript and concluded that the remaining minor concerns have been sufficiently addressed.

I am thus pleased to inform you that your manuscript has been accepted for publication in the EMBO Journal.

Related I would like to hereby ask your consent on keeping the referee figures included in this file.

On a different note, I would like to alert you that EMBO Press offers a format for a video-synopsis of work published with us, which essentially is a short, author-generated film explaining the core findings in hand drawings, and, as we believe, can be very useful to increase visibility of the work. Please see the following link for representative examples and their integration into the article web page:

<https://www.embopress.org/doi/full/10.15252/emj.2019103932>

Best regards,

Daniel Klimmeck

Daniel Klimmeck, PhD
Senior Editor
The EMBO Journal
EMBO

Postfach 1022-40
Meyerhofstrasse 1
D-69117 Heidelberg
contact@embojournal.org
Submit at: <http://emboj.msubmit.net>
